# VISIOMATH: BENCHMARKING FIGURE-BASED MATHEMATICAL REASONING IN LMMS

**Can Li**[1,2,3]    **Ying Liu**[1,2,3]    **Ting Zhang**[1,2,3†]    **Mei Wang**[1,2,3]    **Hua Huang**[1,2,3]

[1] School of Artificial Intelligence, Beijing Normal University
[2] Beijing Key Laboratory of Artificial Intelligence for Education
[3] Engineering Research Center of Intelligent Technology and Educational Application, Ministry of Education

## ABSTRACT

Large multimodal models have achieved remarkable progress in integrating vision and language, enabling strong performance across perception, reasoning, and domain-specific tasks. However, their capacity to reason over multiple, visually similar inputs remains insufficiently explored. Such fine-grained comparative reasoning is central to real-world tasks, especially in mathematics and education, where learners must often distinguish between nearly identical diagrams to identify correct solutions. To address this gap, we present VisioMath, a curated benchmark of 1,800 high-quality K–12 mathematics problems in which all candidate answers are diagrams with subtle visual similarities. A comprehensive evaluation of state-of-the-art LMMs, covering both leading closed-source systems and widely adopted open-source models, reveals a consistent decline in accuracy as inter-image similarity increases. Analysis indicates that the dominant failure mode stems from image–text misalignment: rather than grounding reasoning in textual cues, models often resort to shallow positional heuristics, resulting in systematic errors. We further explore three alignment-oriented strategies, spanning training-free approaches and finetuning, and achieve substantial accuracy gains. We hope that VisioMath will serve as a rigorous benchmark and catalyst for developing LMMs toward deeper diagram understanding, precise comparative reasoning, and grounded multi-image–text integration. The code and dataset are available at `https://github.com/Nefefilibata/VisioMath`.

## 1 INTRODUCTION

In recent years, Large Multimodal Models (LMMs) (Chen et al., 2025; OpenAI, 2024; Team, 2024a; Wang et al., 2024c; Wu et al., 2024b) have achieved remarkable success across various multimodal tasks. This surge in capabilities is largely attributed to the availability of massive, high-quality vision-and-language datasets (Chen et al., 2023; He et al., 2023; Kuznetsova et al., 2020; Singla et al., 2024), which enable the training of increasingly capable models. By jointly modeling image and text modalities, LMMs enable seamless cross-modal reasoning, allowing for the interpretation of complex visual scenes in natural language and vice versa. This integration not only enhances basic perceptual capabilities but also supports high-level cognitive tasks such as visual recognition (Chen et al., 2024b; Huang et al., 2024; Wang et al., 2024d), logical reasoning (Wang et al., 2024e; Wu et al., 2024a; Xiao et al., 2024), and context understanding (Zhang et al., 2024a).

With the rapid development of LMMs, designing holistic benchmarks is essential for systematically investigating the capabilities and limitations of these models. Numerous evaluation benchmarks have been proposed, targeting different aspects of LMM performance, including perception, reasoning, domain-specific tasks, hallucination, and multimodal integration (Huang & Zhang, 2024; Li et al., 2024c). Among these, multimodal reasoning ability, particularly mathematical reasoning that requires integrating visual and textual information, has always been a central focus. This form of reasoning presents distinct challenges, requiring not only the understanding of mathematical semantics in text but also the accurate interpretation and synthesis of visual representations.

---

[†]Corresponding author.

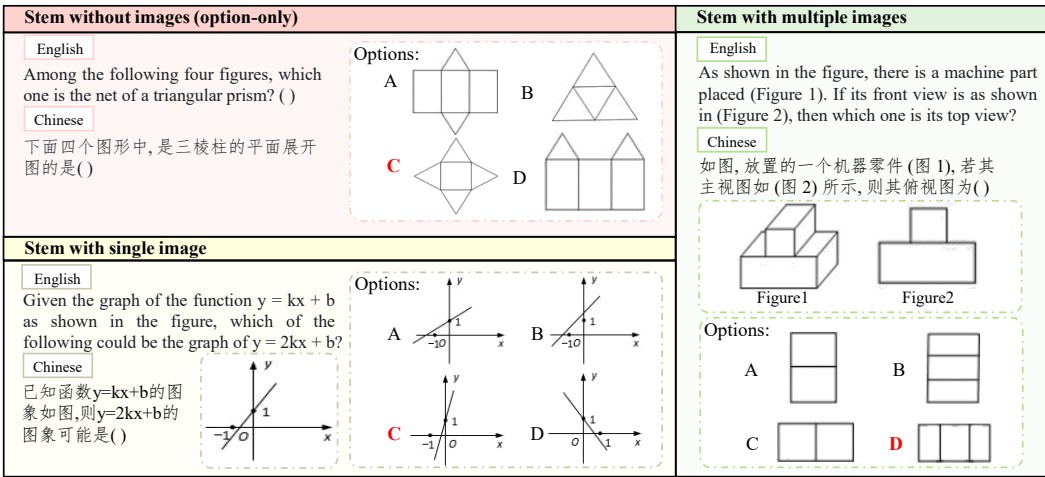

Figure 1: Illustrating examples in our VisioMath dataset, in which data samples consist of visual answer options exhibiting high visual similarity, and the stem may appear with or without images.

To evaluate multimodal reasoning capabilities, various multimodal mathematical reasoning benchmarks have been introduced (Lu et al., 2024; Zhang et al., 2024b; Wang et al., 2024b). These benchmarks can be broadly divided into two categories. The first involves single-image scenario, where each problem is paired with a single diagram that supplements the text. While effective for assessing basic multimodal understanding, these setups are limited in capturing the complexity of real-world visual reasoning, as a single image often lacks the richness and inter-image dependencies needed for higher-order comprehension. In response, recent studies have shifted toward the second category: multi-image scenario. These tasks require reasoning across problems with multiple visual inputs. This paper also investigates on multi-image scenario with a particular emphasis on a specific and underexplored setting: reasoning over multiple highly similar images.

In this paper, we examine a distinct class of multimodal benchmarks in which all answer choices are presented as images. Our motivation arises from the observation that many real-world mathematical problems, especially in educational settings, present options as diagrams (e.g. geometric figures). Addressing such problems involves more than visual recognition; it necessitates comparison of visually similar structures and reasoning about subtle symbolic differences. While recent benchmarks such as CMM-Math-test (Liu et al., 2024b), MathVerse-mv (Li et al., 2024b), and MV-Math (Wang et al., 2025b) have advanced the evaluation of multimodal reasoning by introducing multi-image questions, they often overlook a crucial aspect where reasoning must be grounded in perceptually similar visual features. Our work aims to address this gap and thereby provide an evaluation perspective that specifically targets LMMs' reasoning across closely resembling images.

To achieve that, we introduce VisioMath, a novel benchmark comprising 1,800 meticulously curated, high-quality mathematics problems. The dataset spans a broad spectrum of K–12 mathematics topics, including geometry, algebraic visualizations, numerical comparisons, and functional pattern recognition, thereby capturing the diversity of real-world curricula. The focus on K–12 mathematics is deliberate: fine-grained comparative reasoning is prevalent in this domain, where students must distinguish nearly identical diagrams to identify correct solutions. This makes VisioMath not only an ideal benchmark for evaluating LMMs' visual-textual grounding capabilities but also directly relevant for improving their potential to support K–12 tutoring and educational applications. Specifically, each problem features diagrammatic answer options, with approximately 50% also incorporating at least one image in the question stem to provide essential visual context. To ensure accuracy and reliability, each question has been independently annotated and cross-validated by at least two expert annotators. To reduce answer-choice bias in LMMs, we enforce a uniform distribution across the four multiple-choice options (A, B, C, D). As shown in Figure 1, each answer option is a distinct diagram differing subtly from the others, requiring fine-grained visual discrimination.

We conduct a comprehensive evaluation on the VisioMath benchmark. Our study encompassed a diverse set of LMMs across various model families and scales, including state-of-the-art closed-

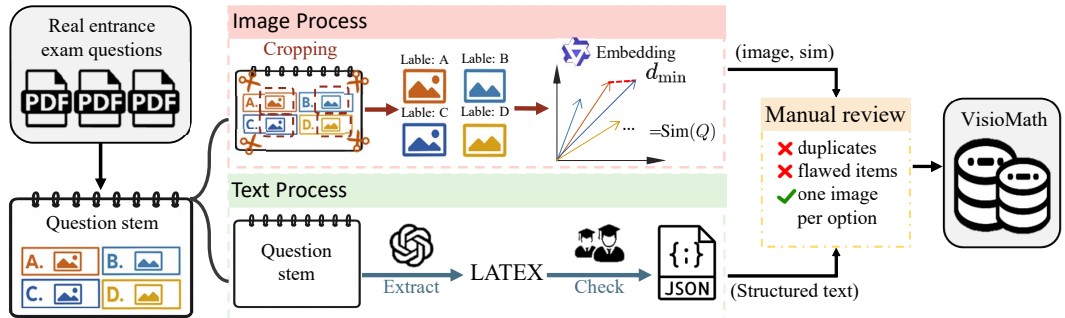

Figure 2: Data processing pipeline of VISIOMATH, including text extraction and verification, image cropping, and integration of visual similarity information to construct the final dataset.

source models such as GPT-4.1 (OpenAI, 2025) and Gemini2.5 Pro (Comanici et al., 2025), as well as prominent open-source models like Qwen2.5-VL (Bai et al., 2025). These models span different input paradigms, with some restricted to single-image input and others capable of processing multiple images simultaneously. We perform a detailed error analysis and find that image–text misalignment accounts for the largest proportion. This highlight a fundamental overlooked limitation in current LMMs: their inability to reliably establish fine-grained correspondences between multiple images and distinct textual inputs. In tasks such as image–option problems, where each figure must be uniquely paired with a specific textual option, LMMs often fail to preserve these one-to-one mappings. This weakness indicates that, although LMMs excel in single-image reasoning and holistic multimodal understanding, they remain inadequate when tasks demand precise cross-modal alignment across multiple visual–text pairs.

We further explore three complementary strategies aimed at mitigating image–text misalignment and enhancing multi-image reasoning: consolidating multiple images into a single layout, establishing explicit visual–textual anchors, and fine-tuning with an alignment-oriented multi-image chain-of-thought dataset. Notably, such limited Chain-of-Thought(CoT) fine-tuning data yields a substantial accuracy gain (+12.6%), illustrating the critical role of explicit visual–textual alignment in enabling effective multi-image reasoning. We hope our work will motivate more systematic exploration of methods for enhancing multi-image–text alignment in complex reasoning tasks.

In summary, our key contributions are:

- **VisioMath Benchmark.** We introduce VisioMath, the first benchmark specifically designed for image-option mathematical reasoning. It bridges the gap between traditional multimodal visual question-answering benchmarks, providing a rigorous testbed for evaluating LMMs' diagram understanding and fine-grained visual reasoning.

- **Comprehensive Evaluation.** We systematically evaluate a wide range of state-of-the-art LMMs, including GPT-4.1 and Gemini2.5 pro, and reveal that even top-performing models struggle with reasoning over visually similar answer options, highlighting a critical limitation when dealing with complex reasoning requiring multi-image-text alignment.

- **Analytical Strategies.** We perform detailed error analyses to identify key failure modes, design controlled experiments to validate the critical limitation, and introduce alignment-focused strategies that substantially improve figure-based reasoning performance.

## 2 VISIOMATH

**Motivation.** In mathematics education, multiple-choice questions with diagrammatic answer options are pervasive. These diagrams often exhibit high visual similarity, differing only in subtle geometric structures or functional curves. Humans can reliably leverage such fine-grained differences through prior knowledge and structured reasoning. In contrast, LMMs typically rely on superficial embedding similarity, making it difficult to discriminate between nearly identical options.

Routine for students, this setting remains unexpectedly challenging for LMMs. As illustrated in Figure 1, the four candidate diagrams share almost identical visual styles, yet solving the problem

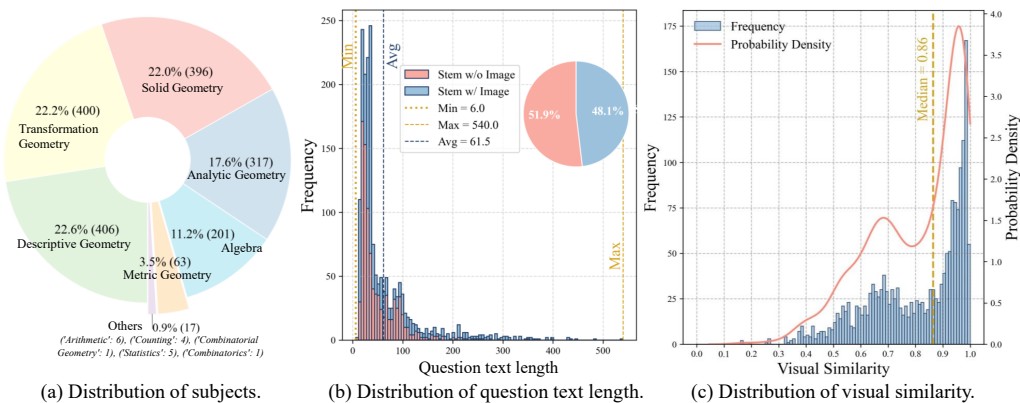

(a) Distribution of subjects.    (b) Distribution of question text length.    (c) Distribution of visual similarity.

Figure 3: Detailed statistics of the VisioMath dataset. The figure shows distributions of (a) subject, (b) question text length and (c) visual similarity, highlighting both textual and visual characteristics.

requires aligning the textual description with precise visual interpretation. To capture this ubiquitous but underexplored scenario, we introduce VisioMath, a benchmark explicitly designed to evaluate LMMs' reasoning ability over multiple highly similar image options in mathematics.

## 2.1 BENCHMARK CONSTRUCTION

Building on the motivation introduced above, VisioMath is constructed to faithfully reproduce exam-like scenarios where reasoning hinges on subtle visual distinctions. To this end, during construction we follow three design principles, *representativity, reliability, and high visual similarity*. The overall construction pipeline is illustrated in Figure 2.

**Representativity.** VisioMath contains 1,800 multiple-choice questions with 8,070 diagrammatic options, collected from Chinese high school and college entrance examinations administered between 2002 and 2023. Using real exam items ensures external validity: the benchmark directly reflects the types of problems students actually face, and its results can generalize to real educational scenarios. Each problem is paired with option diagrams as well as stem diagrams (average 4.48), and we intentionally balanced the correct answer distribution across A–D (24–26% each) to eliminate positional bias. The question text length average 61.5 tokens, reflecting moderate linguistic complexity as shown in Figure 3 (b). We also present the distribution of subject areas across the dataset in Figure 3 (a), offering an integrated overview of the benchmark's coverage.

**Reliability.** To ensure that evaluation results reflect genuine reasoning ability rather than spurious cues, we standardize and curate all samples. Question texts were digitized into a consistent JSON format, where mathematical expressions were transcribed into LaTeX to guarantee uniform parsing. Answer diagrams were carefully cropped from PDFs to enforce a strict one image per option rule, preventing layout or formatting artifacts from providing shortcuts. Finally, all items underwent manual review to eliminate duplicates, low-quality images, and conceptually flawed questions. These steps establish a dataset that is reliable to faithfully evaluate the visual reasoning ability of LMMs.

**High Visual Similarity.** A distinctive property of VisioMath lies in its *systematic quantification of visual similarity* among answer options. For each question $Q$, option images $x_i$ are encoded using the Qwen multimodal-embedding-v1 model, and the question-level visual similarity is then defined as the minimum pairwise cosine similarity across all encoded images:

$$\text{Sim}(Q) = \min_{i \neq j} \cos\big(f(x_i), f(x_j)\big), \tag{1}$$

where $f(\cdot)$ denotes the image embedding encoder.

As illustrated in Figure 3 (c), a large proportion of VisioMath problems contain *highly similar options*, creating fine-grained distinctions that are especially challenging for LMMs. Importantly, we preserve the full spectrum of similarity levels rather than filtering out low-similarity cases, so that performance can be systematically compared under different similarity regimes.

Table 1: Comparison between VisioMath and existing evaluation datasets. Here, EN and CN denote English and Chinese, respectively; FO refers to figure-based options; and AvgImg indicates the average number of images for each problem.

| Datasets | Multi-image problem | Language | #Problems (FO) | #Problems | #Images | AvgImg |
|---|---|---|---|---|---|---|
| We-Math (Qiao et al., 2024) | ✗ | EN | – | 6500 | 6500 | 1 |
| MMMU-Math (Yue et al., 2024) | ✗ | EN | – | 540 | 540 | 1 |
| Math-Vista (Lu et al., 2024) | ✗ | EN | – | 6141 | 6141 | 1 |
| Math-Verse (Zhang et al., 2024b) | ✗ | EN | – | 2612 | 2612 | 1 |
| Math-Vision (Wang et al., 2024b) | ✗ | EN | – | 3040 | 3040 | 1 |
| MM-Math (Sun et al., 2024) | ✗ | EN | – | 5,929 | 5,929 | 1 |
| CMMU-MATH (He et al., 2024) | ✗ | CN | – | 778 | 778 | 1 |
| MathExplain (Park et al., 2025) | ✗ | EN | – | 997 | 997 | 1 |
| MathGlance (Sun et al., 2025) | ✗ | EN | – | 1,609 | 1,609 | 1 |
| Gaokao-MM-Math (Zong & Qiu, 2024) | ✔ | CN | 17 | 80 | 142 | 1.78 |
| CMM-Math-test (Liu et al., 2024b) | ✔ | CN | 245 | 5821 | 3794 | 2.26 |
| MathVerse-mv (Li et al., 2024b) | ✔ | EN | 0 | 788 | 6304 | 8 |
| MV-Math (Wang et al., 2025b) | ✔ | CN,EN | 595 | 2009 | 6061 | 3.02 |
| VisioMath (Ours) | ✔ | CN,EN | 1800 | 1800 | 8070 | 4.48 |

## 2.2 BENCHMARK ANALYSIS

**Unique Challenges.** VisioMath introduces a set of unique challenges that distinguish it from existing multimodal benchmarks. Unlike conventional tasks that pair a single image with text, VisioMath requires reasoning across multiple diagrammatic options simultaneously, transforming the problem into one of comparative visual reasoning that mirrors authentic exam scenarios. Moreover, the benchmark faithfully preserves the presence of highly similar distractors, where candidate diagrams differ only in subtle geometric or symbolic details, thereby testing models' capacity for fine-grained perceptual discrimination. Finally, effective problem solving in VisioMath demands precise text–visual alignment, as models must ground linguistic conditions such as symmetry, monotonicity, or functional transformations in the correct image choice. Collectively, these characteristics elevate VisioMath from simple image recognition to a rigorous evaluation of figure-based visual reasoning.

**Benchmark Comparison.** We compare VisioMath with prior multimodal math benchmarks in Table 1. Most existing datasets adopt a single-image setting with textual answer options (e.g., We-Math, MMMU, Math-Vista, Math-Verse, Math-Vision). Multi-image formats are rare, and when present, image-based options are either absent or inconsistently represented. For instance, MathVerse-mv includes multiple images but no image in options. CMM-Math-test and MV-Math contain some image-based options, yet many are embedded in composite layouts rather than provided as independent visual elements. VisioMath, in contrast, explicitly structures answer options as collections of distinct and semantically meaningful images, thereby supporting a more nuanced evaluation of fine-grained visual mathematical reasoning.

## 3 EXPERIMENT

**Setup.** To comprehensively evaluate the performance of LMMs in handling complex visual inputs, we select a diverse set of models across different accessibility types and input configurations. Specifically, we include closed-source LMMs representing the current state-of-the-art in commercial multimodal systems. In addition, we conduct experiments on open-source LMMs that explicitly support multi-image inputs with various model sizes. This broad coverage ensures a representative analysis across model capacities and architectures. Finally, we include LMMs that have been specifically trained on mathematical corpora and optimized for mathematical QA tasks. All LMMs are evaluated under zero-shot setting to ensure a fair and consistent comparison of their generalization capabilities. More details are provided in Appendix A.

## 3.1 RESULTS

Table 2 reports the comparative performance of various LMMs on VisioMath benchmark, with results disaggregated by the ground-truth (GT) answer position (A–D). The evaluation considers two distinct conditions: (i) question stems presented without images and (ii) question stems accompanied by images. For each condition, we provide both average accuracy and per-option performance.

Table 2: Performance comparison on VisioMath with results categorized based on GT position.

| Models \GT position | Avg | Question stem w/o images | | | | | Question stem with images | | | | |
|---|---|---|---|---|---|---|---|---|---|---|---|
| | | Avg | A | B | C | D | Avg | A | B | C | D |
| Human | 91.3 | 92.3 | 92.5 | 95.6 | 93.8 | 88.5 | 89.7 | 94.4 | 87.6 | 87.5 | 88.0 |
| Random | 25.6 | 25.4 | 24.0 | 25.6 | 23.0 | 28.6 | 26.0 | 22.8 | 27.6 | 28.4 | 25.6 |
| *Closed-source LMMs* | | | | | | | | | | | |
| QwenVL-max (Bai et al., 2023) | 44.1 | 53.4 | 35.2 | 62.6 | 62.5 | 50.2 | 34.1 | 31.1 | 34.1 | 32.8 | 38.6 |
| GPT4.1 (OpenAI, 2025) | 52.6 | 61.6 | 72.4 | 59.9 | 60.2 | 56.1 | 42.8 | 54.8 | 39.3 | 43.7 | 31.9 |
| Seed1.6-Thinking (ByteDance, 2024) | 72.3 | 85.7 | 90.3 | 87.2 | 82.4 | 83.9 | 58.0 | 71.8 | 53.7 | 44.6 | 59.4 |
| Gemini 2.5 Pro (Comanici et al., 2025) | **80.9** | **86.2** | 89.2 | 84.6 | **85.2** | **86.3** | **75.2** | **78.8** | **77.6** | **75.0** | **68.6** |
| *Open-source LMMs (multi-image input)* | | | | | | | | | | | |
| InternVL2.5-2B (Chen et al., 2024a) | 24.6 | 27.1 | 12.8 | 25.5 | 36.3 | 30.2 | 21.9 | 10.3 | 26.2 | **38.2** | 15.0 |
| Qwen2.5-VL-3B-instruct (Bai et al., 2025) | 25.4 | 26.1 | 51.0 | 40.5 | 14.5 | 5.9 | 24.7 | 18.3 | **70.1** | 5.4 | 4.3 |
| R1-Onevison-7B (Yang et al., 2025) | 29.6 | 35.0 | 38.8 | 37.4 | 34.8 | 30.2 | 23.7 | 22.0 | 32.2 | 28.9 | 11.6 |
| Qwen2.5-VL-7B-instruct (Bai et al., 2025) | 32.7 | 39.5 | 30.1 | 58.1 | 39.8 | 29.8 | 25.3 | 8.7 | 28.5 | 32.4 | 34.3 |
| Gemma3-27B (Team, 2025b) | 35.3 | 43.7 | 67.9 | 40.1 | 33.6 | 38.4 | 26.2 | 40.2 | 24.8 | 12.3 | 25.1 |
| Vision-R1-7B (Huang et al., 2025) | 36.7 | 43.7 | 47.4 | 57.3 | 38.7 | 33.7 | 29.2 | 24.5 | 52.3 | 29.4 | 10.6 |
| Qwen2.5-VL-72B-instruct (Bai et al., 2025) | 43.7 | 53.5 | 36.2 | 63.9 | 61.3 | 49.8 | 33.0 | 29.9 | 37.8 | 29.9 | **35.2** |
| GLM-4.5V (Team et al., 2025) | **53.7** | **69.1** | **71.9** | **75.8** | **68.4** | **61.2** | **37.2** | **46.5** | 42.5 | 31.4 | 26.6 |
| *Open-Source LMMs (math-oriented)* | | | | | | | | | | | |
| MM-PRM-8B (Du et al., 2025) | 31.7 | 38.4 | 28.1 | 43.2 | 44.9 | 35.7 | 24.4 | 10.8 | **41.6** | 35.3 | 11.6 |
| MM-Eureka-7B (Meng et al., 2025) | 37.9 | 50.9 | **36.2** | **62.1** | 52.7 | 50.1 | 24.0 | **21.1** | 22.4 | 27.4 | **25.6** |
| MM-Eureka-7B-CPGD (Liu et al., 2025) | **39.3** | **51.0** | 33.2 | 54.2 | **61.3** | **51.4** | **26.9** | 16.2 | 29.9 | **39.7** | 23.7 |

Table 3 further details the accuracy of LMMs on subsets of the dataset stratified by image similarity levels. The dataset is divided into quartiles based on the degree of visual similarity between images within each question, and model performance is reported separately for each quartile. This analysis aims to evaluate models' fine-grained reasoning capabilities under varying visual similarities.

Based on these results, we have following observations.

**Observation 1** (*Question stems containing images pose greater challenges for LMMs*). As shown in Table 2, most LMMs demonstrate noticeably lower performance on questions whose stems include images compared to those with text-only stems, which is a trend consistent across nearly all positions. This observation suggests that the inclusion of images in the question stem significantly increases the complexity of the visual reasoning task. Specifically, when both the stem and the options involve visual content, LMMs are required to process and integrate multiple sources of visual information, which likely imposes a higher cognitive load on the model. This indicates that current LMMs still struggle with multi-image reasoning scenarios and highlights a potential bottleneck in their capacity for holistic visual understanding.

**Observation 2** (*Performance degrades under high visual similarity*). LMMs exhibit performance degradation on tasks involving high inter-image similarity, as shown in Table 3. For instance, Doubao-1.5-Vision-Pro achieves 74.9% accuracy in the quartile with the lowest similarity, but this drops to 62.0% in the highest-similarity quartile, a 12.9% decline. This performance gap stems from the increased demands for fine-grained cross-image associative reasoning, which current LMMs insufficiently support due to limited visual granularity and reasoning capabilities. Notably, LMMs exhibit strong performance correlations across similarity quartiles: models performing well in low-similarity scenarios tend to retain relative strength under high similarity.

**Observation 3** (*Distinct failure modes in Human and LMMs*). As shown in Table 3, human performance moderately decreases as visual similarity among candidate diagrams increases, confirming that higher similarity introduces additional perceptual challenges. Second, beyond a certain similarity threshold, the accuracy plateaus, suggesting that errors at this stage are driven more by conceptual understanding than by perceptual similarity. This suggests that while high similarity increases perceptual load, humans can still reliably distinguish fine-grained differences through careful observation. In contrast, LMMs frequently fail on perceptually trivial distinctions that humans rarely confuse, as illustrated in the error analysis Figure 9. This disparity indicates that current model failures stem largely from inadequate visual-textual alignment rather than a lack of reasoning depth.

## 3.2 ANALYSIS

**Error Categorization.** We conduct a systematic error analysis of GLM4.5V to better understand the limitations of LMMs on VisioMath. From the model outputs, we randomly sample 50 erro-

Table 3: Performance comparison on VisioMath with results categorized based on image similarity.

| Models \ Image similarity | Avg | [0.16,0.68] | (0.68,0.87] | (0.87,0.96] | (0.96,1] |
|---|---|---|---|---|---|
| Human | 91.3 | 95.7 | 91.2 | 87.6 | 89.0 |
| Random | 25.6 | 23.6 | 24.4 | 27.8 | 27.1 |
| *Closed-source LMMs* | | | | | |
| QwenVL-max (Bai et al., 2023) | 44.1 | 47.3 | 50.2 | 41.3 | 37.6 |
| GPT-4.1 (OpenAI, 2025) | 52.6 | 65.8 | 56.4 | 42.9 | 45.1 |
| Seed1.6-Thinking (ByteDance, 2024) | 72.3 | 82.4 | 74.2 | 66.2 | 66.4 |
| Gemini 2.5 Pro (Comanici et al., 2025) | **80.9** | **86.2** | **83.8** | **76.7** | **76.9** |
| *Open-source LMMs (multi-image input)* | | | | | |
| InternVL2.5-2B (Chen et al., 2024a) | 24.6 | 24.2 | 28.9 | 22.7 | 22.7 |
| Qwen2.5-VL-3B-instruct (Bai et al., 2025) | 25.4 | 26.7 | 27.6 | 24.4 | 22.9 |
| R1-Onevison-7B (Yang et al., 2025) | 29.6 | 21.9 | 32.2 | 28.9 | 11.6 |
| Qwen2.5-VL-7B-instruct (Bai et al., 2025) | 32.7 | 33.6 | 37.8 | 29.8 | 29.6 |
| Gemma3-27B (Team, 2025b) | 35.3 | 43.3 | 41.2 | 29.6 | 26.4 |
| Vision-R1-7B (Huang et al., 2025) | 36.7 | 46.7 | 38.9 | 30.4 | 30.9 |
| Qwen2.5-VL-72B-instruct (Bai et al., 2025) | 43.7 | 47.1 | 50.8 | 38.0 | 38.7 |
| GLM-4.5V (Team et al., 2025) | **53.7** | **68.7** | **59.3** | **44.2** | **44.7** |
| *Open-source LMMs (math-oriented)* | | | | | |
| MM-PRM-8B (Du et al., 2025) | 31.7 | 37.6 | 37.1 | 26.9 | 25.1 |
| MM-Eureka-7B (Meng et al., 2025) | 37.9 | 45.6 | 44.0 | 29.1 | **33.1** |
| MM-Eureka-7B-CPGD (Liu et al., 2025) | **39.4** | **47.8** | **46.0** | **30.9** | 32.9 |

neous cases and manually inspect their characteristics, and we categorize the errors into four types, with their proportions illustrated in Figure 5 (a) (mode details in Appendix D). Among the identified categories, *image–text misalignment* account for the largest shares, representing 36% of the errors. Compared to single-image datasets such as MATH-Vision, these proportions are significantly higher. This finding highlights that reasoning over multiple visual contexts introduces substantial challenges, particularly in maintaining consistent semantic alignment across both images and text.

**Effect of Option Shuffling.** The image–text misalignment errors suggest that current LMMs rely heavily on heuristic positional correspondences between options and images. To investigate this, we conducted a controlled shuffling experiment: the image order was kept unchanged, while the textual references to the options were permuted. For example, the original prompt "The last four pictures are respectively the pictures for options A, B, C, and D" was modified to "The last four pictures are respectively the pictures for options B, C, D, and A," with the ground-truth answers adjusted accordingly. By keeping the image order constant, we isolate the effect of image order on performance. Results shown in Figure 5 (b) suggest a consistent and clear decline under this manipulation. For instance, Gemini 2.5 Pro's accuracy dropped from 80.9% to 72.2% (–8.7%). These findings indicate that existing LMMs struggle to robustly capture and align semantic correspondences between textual options and visual content, highlighting the need for improved cross-modal alignment mechanisms in multi-image reasoning tasks.

**Rationale for Visual Similarity Metric.** We define the visual similarity of a question, $Sim(Q)$, using the minimum pairwise cosine similarity ($S_{min}$) computed by Qwen multimodal-embedding-v1. Here we study the effect of the aggregation strategy and the visual encoders.

*The Aggregation Strategy.* We use $S_{min}$ to establish a strict lower bound for visual discrimination difficulty. Two alternative measures are the mean similarity and the median similarity. Comparing $S_{min}$ with mean and median similarities reveals highly aligned quartile boundaries (Figure 4 (a)) and consistent model performance trends—specifically, a uniform accuracy drop in the highest similarity quartile (Q4) across all methods (Table 4). This confirms $S_{min}$ is a statistically robust metric for structural dataset grouping.

*The Visual Encoders.* Unlike CLIP (optimized for natural images) and BLIP (general QA), Qwen is trained extensively on diagrammatic reasoning, making it uniquely suited for capturing fine-grained semantics in math diagrams. Nearest neighbor retrieval experiments (Figure 4 (b)) validate this choice: Qwen maintains strict geometric and topological consistency (e.g., slope, intercept). Conversely, BLIP suffers from geometric drift after the top-1 result, and CLIP prioritizes high-level semantics over precise visual topology, rendering both unsuitable for fine-grained differentiation.

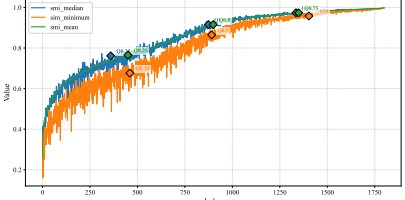 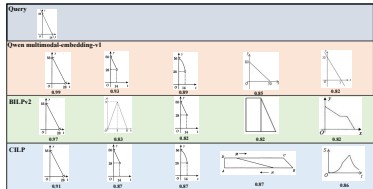

(a) Score distribution of minimum, mean, and median similarity across all problems.

(b) Top-5 retrieval results with Qwen multimodal-embedding-v1, BLIPv2, CLIP.

Figure 4: Visual similarity analysis: (a) distribution of different similarity metrics; (b) retrieval results using different visual encoders.

Table 4: Performance comparison on VisioMath with results categorized based on image similarity quartiles (Q1–Q4) defined by Minimum (min), Mean (mean), and Median (med) aggregation.

| Models \ Image similarity | Q1 | | | Q2 | | | Q3 | | | Q4 | | |
|---|---|---|---|---|---|---|---|---|---|---|---|---|
| | min | mean | med | min | mean | med | min | mean | med | min | mean | med |
| QwenVL-plus (Bai et al., 2023) | 33.3 | 35.6 | 35.1 | 37.8 | 35.1 | 35.8 | 32.4 | 32.9 | 32.0 | 28.2 | 28.2 | 28.9 |
| GPT-4o (OpenAI, 2024) | 53.8 | 56.0 | 55.8 | 50.9 | 49.5 | 50.0 | 40.0 | 40.0 | 40.0 | 39.1 | 39.1 | 40.0 |
| Gemini2-flash-thinking (DeepMind, 2025b) | 63.6 | 66.2 | 66.2 | 58.9 | 57.3 | 56.7 | 48.2 | 46.9 | 46.9 | 42.2 | 42.6 | 41.6 |
| Gemini2-flash (DeepMind, 2025a) | 66.7 | 69.1 | 70.4 | 59.8 | 56.8 | 55.1 | 49.3 | 49.1 | 49.5 | 46.2 | 45.5 | 45.5 |
| Doubao-1.5-Vision-pro (Team, 2025a) | 74.9 | 75.8 | 77.1 | 68.2 | 68.2 | 65.7 | 60.4 | 61.1 | 61.1 | 62.0 | 60.4 | 61.5 |
| Seed1.6-Thinking (ByteDance, 2024) | 82.4 | 84.0 | 82.9 | 74.2 | 72.4 | 74.2 | 66.2 | 67.6 | 66.4 | 67.6 | 66.4 | 66.2 |
| Gemini 2.5 Pro (Comanici et al., 2025) | 86.2 | 89.3 | 88.6 | 83.8 | 82.6 | 84.0 | 76.7 | 76.7 | 76.7 | 76.9 | 76.9 | 76.9 |
| Gemma3-27B (Team, 2025b) | 43.3 | 44.0 | 45.5 | 41.2 | 41.5 | 40.0 | 29.6 | 30.4 | 30.4 | 26.4 | 26.2 | 26.2 |
| Qwen2.5-VL-72B-instruct (Bai et al., 2025) | 47.1 | 50.2 | 49.3 | 50.8 | 46.9 | 48.4 | 38.0 | 39.1 | 38.2 | 38.7 | 38.7 | 38.9 |
| GLM-4.5V (Team et al., 2025) | 68.7 | 68.2 | 68.0 | 59.3 | 54.2 | 54.9 | 44.2 | 46.7 | 45.1 | 39.5 | 44.7 | 40.6 |

## 3.3 STRATEGIES FOR PERFORMANCE ENHANCEMENT

Building on the above analysis of LMM limitations, we explore practical strategies to improve multi-image reasoning performance on VisioMath. These strategies fall into two categories: training-free techniques that leverage structural or labeling cues, and a training-based approach that incorporates specialized multi-image reasoning data. Collectively, they demonstrate the potential to mitigate vision–text misalignment and enhance cross-figure reasoning.

**Strategy 1** (*Consolidated single image layout*). We first examine whether providing all visual information in a single spatial layout improves reasoning. Specifically, option images and stem images are concatenated into a composite image. As shown in Figure 5 (b), this structural simplification consistently improves performance, suggesting that LMMs struggle to distribute attention effectively across multiple independent images. For instance, Seed1.6-Thinking achieves an accuracy increase from 72.3% to 78.7% (+6.4%) under this setup. The results indicate that co-locating visual information helps LMMs reason more effectively over multiple images.

**Strategy 2** (*Explicit visual–textual anchors*). In this strategy, each image is directly associated with its corresponding textual label, either through overlaid or embedded annotations. This experimental setting is designed to evaluate whether establishing explicit visual–textual correspondences can enhance disambiguation and support more accurate decision-making. Empirical results shown in Figure 5 (b) demonstrate that this approach yields notable performance gains: for instance, QwenVL-plus improves from 32.9% to 42.7% (+9.8%), whereas Gemini 2.5 Pro shows a smaller but measurable gain of +0.9%. These results indicate that current LMMs continue to struggle with robustly binding textual content to the corresponding visual elements. Importantly, the findings highlight that carefully designed visual–textual anchors can effectively mitigate misalignment errors, offering a practical pathway to improve multimodal reasoning performance.

**Strategy 3** (*Alignment-oriented multi-image chain-of-thought training*). To further enhance reasoning performance, we develop a specialized multi-image CoT dataset explicitly aimed at improving visual–textual alignment across multiple diagrams. Starting from 1,072 multi-image problems collected online, we first employ QwenVL-Max with an image-caption–style prompt to generate preliminary reasoning paths that describe each diagram individually, ensuring localized alignment

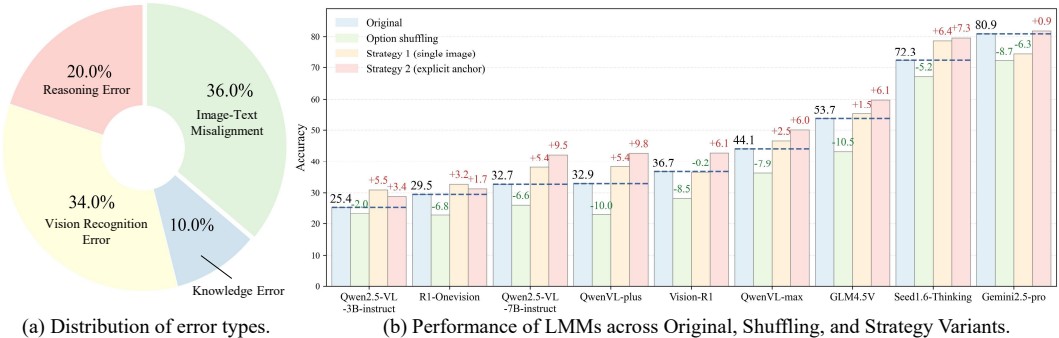

(a) Distribution of error types.  (b) Performance of LMMs across Original, Shuffling, and Strategy Variants.

Figure 5: Illustrating error type distribution and the impact of input data structure on performance.

Table 5: The effect of strategy 3, using alignment-oriented multi-image chain-of-thought fine-tuning.

| Model | Original | Shuffling | Strategy 1 | Strategy 2 | Strategy 3 |
|---|---|---|---|---|---|
| Qwen2.5-VL-3B-instruct (Bai et al., 2025) | 25.4 | 23.4 (-2.0) | 30.9 (+5.5) | 28.8 (+3.4) | 38.0 (+12.6) |
| Qwen2.5-VL-7B-instruct (Bai et al., 2025) | 32.7 | 26.1 (-6.6) | 38.1 (+5.4) | 42.5 (+9.8) | 43.3 (+10.6) |
| Qwen2.5-VL-72B-instruct (Bai et al., 2025) | 43.7 | 35.6 (-8.1) | 47.6 (+3.9) | 50.1 (+6.4) | 51.4 (+7.7) |
| InternVL2.5-2B (Chen et al., 2024a) | 24.6 | 23.1 (-1.5) | 26.3 (+1.7) | 27.2 (+2.6) | 32.2 (+7.6) |

between visual elements and textual explanations. These initial outputs are then refined by DeepSeek V3.1 through a CoT Data Generation Prompt, which enforces step-by-step integration of per-image descriptions into a globally coherent reasoning trajectory, tightly binding visual observations to textual inferences. To guarantee reliability, only samples yielding correct final answers are retained, resulting in 500 high-quality multi-image CoT exemplars with explicit visual–textual anchors. We fine-tuning Qwen2.5-VL with various model sizes and InternVL2.5-2B, and the results in Table 5 show that accuracy increases for all models, despite using only a small set of CoT data. Notably, Qwen2.5-VL-3B increases from 25.4% to 38.0% (+12.6%), surpassing R1-OneVision-7B (29.5%) and Vision-R1 (36.7%). These results highlight that current models are severely constrained by the scarcity of alignment-oriented multi-image CoT training data, and that targeted augmentation with explicit alignment signals can substantially boost figure-based reasoning.

## 4 RELATED WORK

**Multimodal Understanding Benchmarks.** Numerous benchmarks have been introduced to evaluate the capabilities of LMMs. While several multi-image benchmarks, such as Blink (Fu et al., 2024), MUIR (Wang et al., 2024a), and MMIU (Meng et al., 2024), have emerged, they primarily assess basic perceptual abilities—like caption recognition and object counting—falling short in measuring deep reasoning. Simultaneously, evaluating the genuine dependency on visual information remains a critical challenge. Recent studies have moved beyond simple natural image QA to multimodal science problems requiring deeper reasoning capabilities: MMSciBench (Ye et al., 2025) and SeePhys (Xiang et al., 2025) introduce the concept of "Vision-Essential" problems to reveal models' reliance on textual shortcuts, while VisAidMath (Ma et al., 2025) explores active visual reasoning through auxiliary line generation. Unlike prior works that mostly focus on interpreting a single stem or generating aids, VisioMath introduces a challenging benchmark centered on comparative visual discrimination via option-containing images, enabling a more comprehensive assessment of models' fine-grained, multi-image comparative reasoning over visually similar diagrams.

**Mathematical Reasoning Benchmarks.** Various datasets have been proposed to evaluate the mathematical capabilities. Text-based benchmarks such as GSM8K (Cobbe et al., 2021) and MATH (Hendrycks et al., 2021) are widely used. To evaluate mathematical reasoning requiring visual understanding, such as geometry and function graph analysis, several multimodal datasets have recently emerged, for example, Math-Verse (Zhang et al., 2024c), Math-Vista (Lu et al., 2024), and Math-Vision (Wang et al., 2024b). Nonetheless, as LMMs advance in multi-image reasoning, these single-image-focused benchmarks are increasingly inadequate for evaluating their full capabilities.

In response, recent research efforts have begun to explore more complex multi-image reasoning scenarios that better reflect the real-world demands of mathematical problem-solving. Despite recent advances, a key limitation persists: existing multi-image benchmarks such as MathVerse-mv (Li et al., 2024b) and MV-Math (Wang et al., 2025b) often neglect figure-based answer options, which are common in mathematics domain (e.g., geometry problems with diagrammatic options). This gap underscores the need for new benchmarks that support figure-based multi-image reasoning.

## 5 Conclusion and Limitation

We introduce VisioMath, a benchmark designed to evaluate multimodal reasoning in contexts where answer options consist of multiple, highly similar diagrams. This benchmark fills a critical gap in existing evaluation frameworks, which rarely consider the challenges of comparative reasoning across visually confusable candidates. Our experiments reveal that current LMMs perform poorly under these conditions: accuracy declines sharply with increasing inter-image similarity, and frequent errors stem from multi-image–text misalignment. Controlled shuffling experiments further show that many models rely on positional heuristics, exposing fundamental weaknesses in their reasoning mechanisms. We further explore alignment-oriented data augmentation and multi-image CoT finetuning. Results demonstrate that these strategies yield substantial gains, even under limited data regimes, indicating that relatively lightweight interventions can enhance LMMs' capacity for robust visual–textual binding.

While VisioMath provides a rigorous evaluation of multi-image, diagram-based reasoning in mathematics, our current benchmark is limited to K–12 math topics. Extending this benchmark to other domains, such as physics, engineering diagrams, or chemistry molecular structures, would test LMMs' ability to generalize multi-image reasoning across diverse visual-semantic contexts.

## 6 Broader Impact

VisioMath highlights critical limitations in current LMMs, particularly in fine-grained visual–text alignment and figure-based visual reasoning. By providing a targeted evaluation platform, it can guide the development of more accurate multimodal models, benefiting educational applications, intelligent tutoring systems, and diagram understanding in STEM disciplines. However, as with any benchmark, there is a risk of overfitting models to its specific structures; care must be taken to ensure that improvements reflect genuine reasoning capabilities rather than dataset-specific heuristics. Overall, we envision VisioMath supporting both model development and pedagogical research, fostering AI systems that can more effectively interpret and reason over complex visual information.

### Acknowledgment

This work was jointly supported by the National Natural Science Foundation of China (62437001), the Fundamental Research Funds for the Central Universities (2253500001, 2243100004).

### Ethics statement

This research is based on a dataset compiled from publicly available papers from Chinese high school and college entrance examinations administered between 2002 and 2023. All data were sourced and processed in compliance with applicable laws and institutional regulations. During the curation process, we implemented a systematic filtering protocol to identify and remove any potentially harmful, offensive, or otherwise inappropriate content. Consequently, the final dataset used in this work is considered ethically sound and suitable for academic research.

### Reproducibility statement

To ensure the reproducibility of our work, we provide comprehensive documentation of our methodology. Section 2.1 details the construction and annotation of our dataset. Appendix A documents the experimental settings, model version, concrete implementations of Strategy 1 and Strategy 2, and all

hyperparameters used for fine-tuning and evaluation. The construction of our CoT reasoning data is described in Appendix B, and the full set of prompts for both generation and evaluation is available in Appendix C. Together, these materials enable independent researchers to replicate our evaluation and reproduce the reported results.

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

## A    EXPERIMENT DETAILS

### A.1    EXPERIMENT SETTING

All experiments were conducted on a Linux server equipped with two NVIDIA H800 GPUs (each with 80GB of memory). The Python version used in the experiments was 3.9.20, while the version of vllm library was 0.8.1, respectively. Each model evaluation was performed in a zero-shot setting with deterministic decoding (temperature=0). Due to frequent updates and improvements, closed-source models often undergo version changes that can significantly impact evaluation results. Even subtle updates may alter model behavior, performance, or prompt adherence. As such, the results reported in this benchmark are tied to the specific versions used during our evaluation. To ensure transparency and reproducibility, Table 6 lists the exact release dates or version identifiers of all closed-source models evaluated in VisioMath. Readers should be aware that performance discrepancies may arise when using newer or older versions of the same models.

Specifically, we employ the same prompt template across all models to eliminate prompt-induced variance, and fix the decoding temperature to 0 to promote deterministic outputs. Accuracy serves as the primary evaluation metric, measuring the proportion of correctly answered instances. We utilize GLM4-Flash (GLM, 2024) to extract the options from the responses generated by LMMs. In scenarios where the model fails to produce a valid answer, i.e., none of the standard options (A, B, C, or D) can be reliably identified, its response is classified as invalid. Such cases are treated as incorrect predictions in the final accuracy computation.

To ensure a fair comparison, we adopted consistent prompting strategies across the three input types: Original, Strategy 1, and Strategy 2. For Strategy 1, we horizontally concatenated all images with zero-padding. For Strategy 2, we extend each option image by adding a 50-pixel-high strip at the bottom, matching the width of the image, and insert the corresponding option letter (A, B, C, or D) within the strip. Examples of these configurations are illustrated in Figure 6. In Strategy 3, we adopted the Supervised Fine-Tuning (SFT) training strategy on QwenVL2.5-3B. Using a single H800 GPU and the ms-swift framework (Zhao et al., 2024), we set the batch size to 2, the learning rate to 1e-4, and the gradient accumulation steps to 4. The training was conducted over 336 steps.

Table 6: Version information or release dates of evaluated closed-source models.

| Model | Version (release date) |
|---|---|
| GPT-4o (OpenAI, 2024) | 2024-11-20 |
| GPT-4.1 (OpenAI, 2025) | 2025-04-14 |
| Gemini2-flash (DeepMind, 2025a) | 2024-12-11 |
| Gemini2-flash-thinking (DeepMind, 2025b) | 2025-01-21 |
| Gemini 2.5 Pro (Comanici et al., 2025) | 2025-06-17 |
| QwenVL-max (Team, 2025c) | 2025-04-08 |
| QwenVL-plus (Team, 2025c) | 2025-01-25 |
| Doubao-1.5-Vision-pro (Team, 2025a) | 2025-03-28 |
| Seed1.6 (ByteDance, 2024) | 2025-08-15 |
| GLM4V-plus (GLM, 2024) | 2025-01-11 |

### A.2    FULL EXPERIMENTAL RESULTS

Due to space limitations in the main text, we report the full evaluation results of various LMMs on the VisioMath benchmark in Table 7 and Table 8. Moreover, we evaluate the adaptability of models not originally designed for multi-image processing. For these models, we implement an image concatenation strategy, in which all images associated with a given question are merged into a single composite image. Despite its straightforwardness, this approach exposes critical limitations. Among the models evaluated, the best performer, LLaVA-v1.6-vicuna-13B achieves only 24.4% accuracy, which is on par with the naive baseline of random guessing. These results underscore a fundamental limitation of single-image LMMs in multi-image contexts: they fail to effectively model relational information across distinct visual inputs. This highlights the need for architectures that explicitly support cross-image representation learning and comparative reasoning.

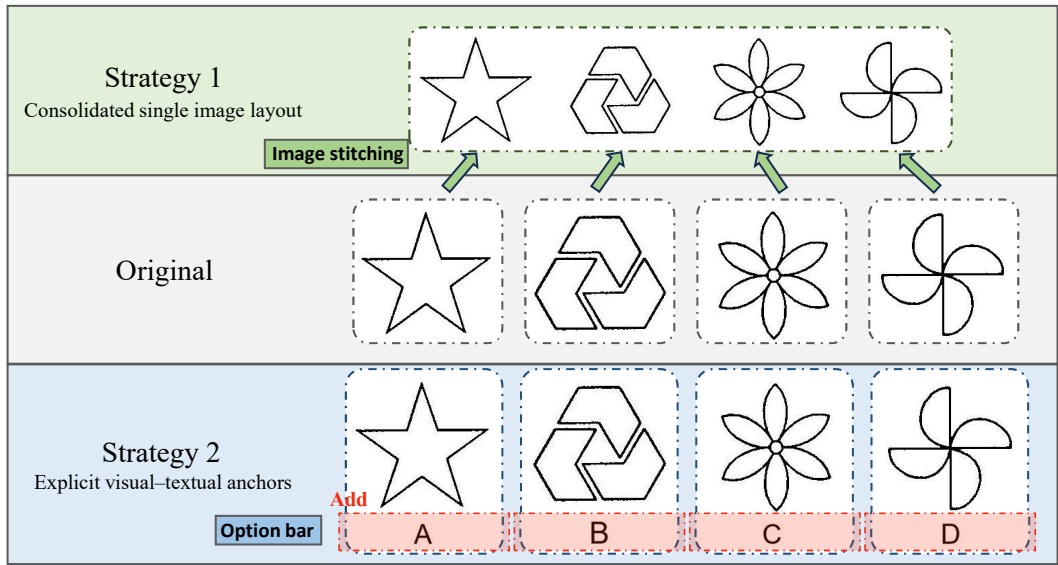

Figure 6: Illustrating the image format of original, strategy 1, and strategy 2 in our experiments.

## B  MULTI-IMAGE CoT FINE-TUNING

This section explains how CoT reasoning data was constructed, including description generation by QwenVL-Max, refinement by DeepSeek3.1, and filtering strategies. We construct a specialized multi-image chain-of-thought (CoT) dataset through a structured three-stage pipeline to enhance model performance.

**Stage 1** (*Problem collection*). We crawled 1,072 mathematical single-choice questions that contain more than four pictures from the internet to serve as the raw problem pool.

**Stage 2** (*CoT Sample Generation*). Initial reasoning paths and descriptive captions are produced for each problem using QwenVL-Max with an Image-Caption–Style Prompt. These outputs, together with the original questions, are then fed into DeepSeek V3.1 via a CoT Data Generation Prompt to generate refined reasoning trajectories and corresponding answers. Samples are subsequently filtered based on answer correctness, resulting in 500 high-quality multi-image CoT examples.

**Stage 3** (*Dataset Expansion*). To increase both the scale and diversity of the dataset, a Option Shuffling Prompt strategy is applied, expanding the dataset from 0.5k to 1.3k samples.

This three-stage pipeline ensures that the final dataset contains both high-quality reasoning examples and sufficient data scale, providing a robust foundation for effective model training.

## C  PROMPT TEMPLATES

We employ the same prompt template across all models to eliminate prompt-induced variance. Specifically, we use five types of system prompts in our paper:

- **Original Answer Prompt**: The baseline system instruction that is uniformly appended to all models prior to evaluation, serving to standardize response format and output scope.

- **Option Shuffling Prompt**: A variant of the Option Shuffling Prompt in which the correspondence between options and images is completely deranged, designed to test and mitigate the model's reliance on positional priors, and used for synthetic data generation.

- **Answer Extraction Prompt**: A prompt used to guide the LLM in extracting and normalizing the final answer from the model's output (e.g., mapping free-form text or reasoning steps to discrete options such as A/B/C/D).

- **Image-Caption–Style Prompt**: A prompt that instructs the MLLM to generate concise, comparable textual descriptions and preliminary analyses for each image, serving as a cross-modal representation bridge.
- **CoT Data Generation Prompt**: A prompt that integrates the question, image captions, and MLLM-provided reasoning trajectories to produce high-quality chain-of-thought rationales and final answers, which can be leveraged for data augmentation and fine-tuning.

The detailed prompt texts are shown in Table 9.

# D  ERROR ANALYSIS

This section presents a detailed analysis of errors, categorizing them into four types, reporting their distributions, and providing representative examples.

**Image-Text Misalignment (36%).** These errors occur when GLM-4.5V fails to correctly capture the semantic correspondence between textual options and visual content. For example, in Figure 7, the model misinterprets the relationship between the image and the answer options, incorrectly treating the reference image as Option A.

**Vision Recognition Error (34%).** Vision recognition errors reflect the model's difficulty in accurately perceiving visual information. As shown in Figure 8, GLM-4.5V fails to correctly interpret the shapes of the unfolded cubes corresponding to Options B and C.

**Reasoning Error (20%).** Reasoning errors arise when GLM-4.5V does not correctly follow logical steps or underlying problem constraints. For instance, in Figure 8, the model incorrectly assumes that the depicted line graph necessarily satisfies the definition of a function.

**Knowledge Error (10%).** Knowledge errors occur when GLM-4.5V lacks relevant domain knowledge or produces outdated/inaccurate information. For example, in Figure 7, the model erroneously interprets the top view of a sphere as a circle with a visible center point.

## D.1  VISION RECOGNITION ERROR ANALYSIS

We further conduct a detailed analysis of vision recognition errors, report their distributions, and provide representative examples.

**Fine-Grained Geometric Perception Error (18%).** Fine-grained geometric perception errors occur when GLM-4.5V struggles to distinguish subtle quantitative differences between highly similar options. For instance, in Figure 9, the model incorrectly distinguishes between Option B and Option C based on the presence of hollow points.

**Spatial Topology & Transformation Error (24%).** These errors occur when GLM-4.5V fails to comprehend the mapping between 2D shapes and 3D objects or understand spatial connectivity. A representative example is shown in Figure 9, where GLM-4.5V fails to correctly interpret the left view of the given solid.

**Spatial Positional Relation Error (22%).** Spatial positional relation errors occur when GLM-4.5V misjudges the relative position of geometric elements within a coordinate system. For example, as shown in Figure 10, the model erroneously assumes that the graph of the function lies entirely above the x-axis.

**Abstract Global Pattern Recognition Error (36%).** Abstract global pattern recognition errors arise when GLM-4.5V identifies local features but fails to integrate them into a coherent global geometric pattern or structural layout. As shown in Figure 10, GLM-4.5V fails to determine whether the figure exhibits axial symmetry.

# E  USE OF LLMS

In this work, large language models (LLMs) were utilized as supporting tools to aid in the revision and polishing of certain text segments during manuscript preparation. All model-generated content was thoroughly examined, revised, and refined by the authors to guarantee accuracy and compliance

with academic writing standards. Notably, the study's conceptual framework, methodological design, data analysis, and interpretation of findings were conducted entirely by the authors, without dependence on automated systems. The authors bear full responsibility for the originality, validity, and integrity of the research presented in this paper.

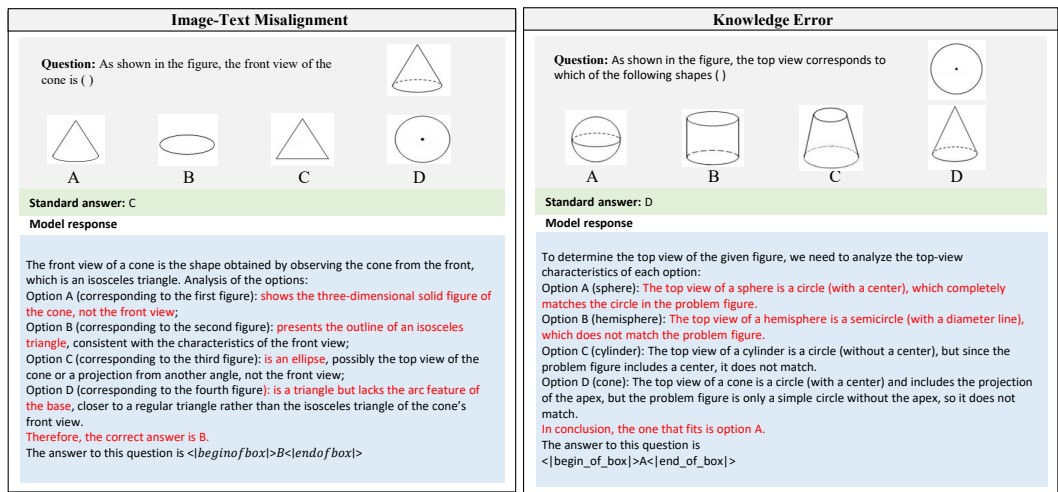

Figure 7: Example error cases for image-text misalignment and knowledge error.

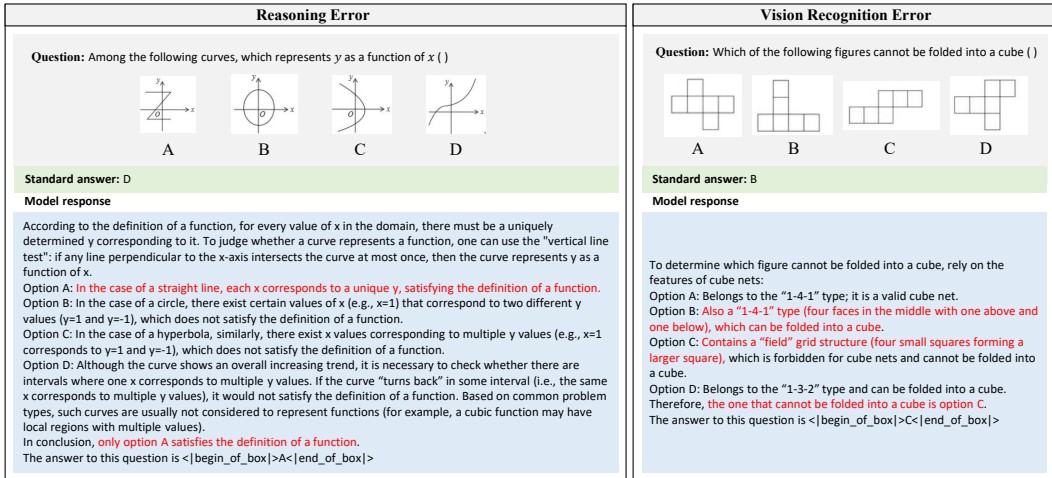

Figure 8: Example error cases for reasoning error and vision recognition error.

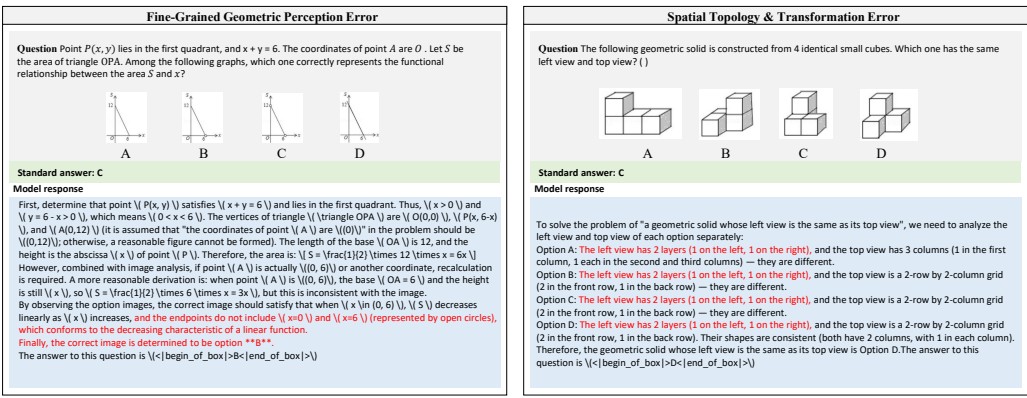

Figure 9: Example error cases for Fine-Grained Geometric Perception Error and Spatial Topology & Transformation Error.

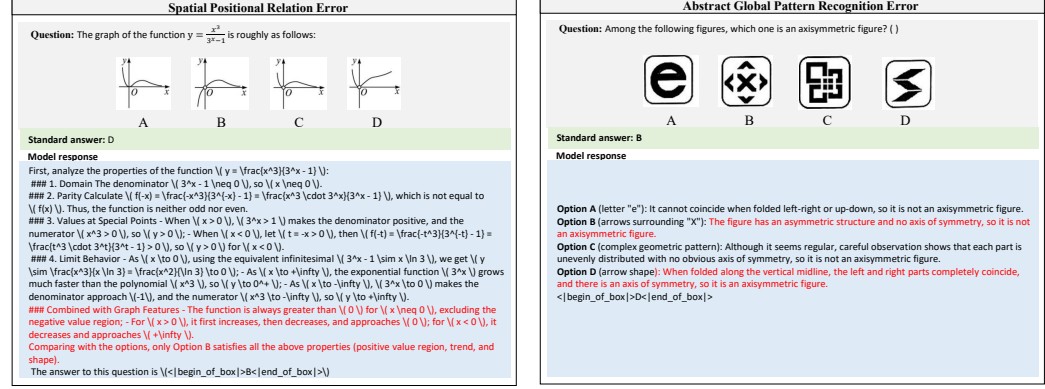

Figure 10: Example error cases for Spatial Positional Relation Error and Abstract Global Pattern Recognition Error.

Table 7: Performance comparison on VisioMath with results categorized based on GT position.

| Models \ GT position | Avg | Question stem w/o images | | | | | Question stem with images | | | | |
|---|---|---|---|---|---|---|---|---|---|---|---|
| | | Avg | A | B | C | D | Avg | A | B | C | D |
| Human | 91.3 | 92.3 | 92.5 | 95.6 | 93.8 | 88.5 | 89.7 | 94.4 | 87.6 | 87.5 | 88.0 |
| Random | 25.6 | 25.4 | 24.0 | 25.6 | 23.0 | 28.6 | 26.0 | 22.8 | 27.6 | 28.4 | 25.6 |
| *Closed-source LMMs* | | | | | | | | | | | |
| GLM4V-plus (GLM, 2024) | 27.9 | 30.2 | 28.1 | 33.5 | 31.6 | 27.5 | 25.4 | 39.4 | 22.9 | 26.0 | 11.1 |
| QwenVL-plus (Bai et al., 2023) | 32.9 | 39.1 | 27.0 | 59.9 | 43.4 | 25.5 | 26.3 | 7.5 | 26.2 | 34.8 | 40.1 |
| QwenVL-max (Bai et al., 2023) | 44.1 | 53.4 | 35.2 | 62.6 | 62.5 | 50.2 | 34.1 | 31.1 | 34.1 | 32.8 | 38.6 |
| GPT-4o (OpenAI, 2024) | 45.9 | 54.7 | 55.6 | 56.4 | 54.7 | 52.5 | 36.5 | 47.3 | 30.4 | 36.3 | 30.4 |
| GPT4.1 (OpenAI, 2025) | 52.6 | 61.6 | 72.4 | 59.9 | 60.2 | 56.1 | 42.8 | 54.8 | 39.3 | 43.7 | 31.9 |
| Gemini2-flash-thinking (DeepMind, 2025b) | 53.2 | 61.2 | 80.6 | 59.9 | 58.6 | 50.3 | 44.6 | 57.3 | 43.0 | 43.1 | 32.9 |
| Gemini2-flash (DeepMind, 2025a) | 55.5 | 65.1 | 78.1 | 59.9 | 65.2 | 59.6 | 45.2 | 57.7 | 34.5 | 38.7 | 47.8 |
| Doubao-1.5-Vision-pro (Team, 2025a) | 66.3 | 75.6 | 78.6 | 78.0 | 76.6 | 70.2 | 56.4 | 73.4 | 55.6 | 48.0 | 45.4 |
| Seed1.6-Thinking (ByteDance, 2024) | 72.3 | 85.7 | **90.3** | **87.2** | 82.4 | 83.9 | 58.0 | 71.8 | 53.7 | 44.6 | 59.4 |
| Gemini 2.5 Pro (Comanici et al., 2025) | **80.9** | **86.2** | 89.2 | 84.6 | **85.2** | **86.3** | **75.2** | **78.8** | **77.6** | **75.0** | **68.6** |
| *Open-source LMMs (multi-image input)* | | | | | | | | | | | |
| deepseekvl2-tiny (Wu et al., 2024b) | 23.5 | 21.6 | 45.9 | 29.1 | 15.2 | 2.7 | 25.6 | 58.5 | 27.1 | 6.9 | 4.3 |
| InternVL2.5-2B-MPO (Chen et al., 2024a) | 23.9 | 24.9 | 15.3 | 26.9 | 33.2 | 22.4 | 22.9 | 13.7 | 28.5 | 36.3 | 14.5 |
| InternVL2.5-2B (Chen et al., 2024a) | 24.6 | 27.1 | 12.8 | 25.5 | 36.3 | 30.2 | 21.9 | 10.3 | 26.2 | 38.2 | 15.0 |
| Llama3.2-11B-Vison (Dubey et al., 2024) | 25.3 | 26.2 | 30.6 | 24.7 | 39.8 | 10.6 | 24.2 | 30.2 | 23.4 | 36.3 | 6.3 |
| Idefics3-8B-llama (Laurençon et al., 2024) | 25.4 | 26.1 | 24.6 | 28.6 | 19.1 | 10.6 | 24.6 | 39.8 | 32.7 | 11.3 | 11.6 |
| Qwen2.5-VL-3B-instruct (Bai et al., 2025) | 25.4 | 26.1 | 51.0 | 40.5 | 14.5 | 5.9 | 24.7 | 18.3 | **70.1** | 5.4 | 4.3 |
| Phi3.5-vision (Abdin et al., 2024) | 25.7 | 25.3 | 73.5 | 22.0 | 14.1 | 2.4 | 26.2 | **78.8** | 13.6 | 3.4 | 0.5 |
| deepseekvl2-small (Wu et al., 2024b) | 26.6 | 32.0 | 42.8 | 28.6 | 30.4 | 28.2 | 20.8 | 38.6 | 12.6 | 12.3 | 14.5 |
| Mantis-8B-Idefics2 (Jiang et al., 2024) | 27.9 | 30.8 | 24.0 | 17.6 | 42.1 | 36.5 | 24.8 | 22.0 | 7.0 | 32.4 | 39.1 |
| InternVL2.5-4B (Chen et al., 2024a) | 28.2 | 30.4 | 23.5 | 37.0 | 34.0 | 26.3 | 25.8 | 22.0 | 36.0 | 34.8 | 10.6 |
| InternVL2.5-4B-MPO (Chen et al., 2024a) | 28.4 | 30.9 | 12.8 | 33.0 | 31.6 | 42.4 | 25.6 | 9.5 | 27.1 | 47.1 | 21.7 |
| MiniCPM-o-2.6 (Abdin et al., 2024) | 29.3 | 34.6 | 40.8 | 36.1 | 31.3 | 31.8 | 23.6 | 36.5 | 23.8 | 15.7 | 15.9 |
| R1-Onevison-7B (Yang et al., 2025) | 29.6 | 35.0 | 38.8 | 37.4 | 34.8 | 30.2 | 23.7 | 22.0 | 32.2 | 28.9 | 11.6 |
| MiniCPM-V-2.6 (Abdin et al., 2024) | 29.7 | 33.0 | 31.6 | 30.0 | 30.9 | 38.4 | 26.1 | 30.0 | 26.2 | 16.7 | 30.9 |
| InternVL2.5-8B (Chen et al., 2024a) | 29.9 | 33.1 | 26.4 | 31.7 | 50.4 | 22.0 | 26.6 | 30.7 | 16.4 | 40.7 | 18.4 |
| InternVL2.5-8B-MPO (Chen et al., 2024a) | 30.9 | 35.9 | 25.0 | 48.5 | 39.5 | 29.4 | 25.5 | 23.7 | 25.2 | 34.3 | 19.3 |
| QvQ-72B-Preview (Team, 2024b) | 30.9 | 36.2 | 45.4 | 36.1 | 44.0 | 31.4 | 25.3 | 31.5 | 28.0 | 26.0 | 14.5 |
| Qwen2-VL-72B-instruct (Wang et al., 2024c) | 31.7 | 38.2 | 15.8 | 29.5 | 78.1 | 23.1 | 24.5 | 2.5 | 9.8 | 83.8 | 6.8 |
| Qwen2.5-VL-7B-instruct (Bai et al., 2025) | 32.7 | 39.5 | 30.1 | 58.1 | 39.8 | 29.8 | 25.3 | 8.7 | 28.5 | 32.4 | 34.3 |
| Gemma3-27B (Team, 2025b) | 35.3 | 43.7 | 67.9 | 40.1 | 33.6 | 38.4 | 26.2 | 40.2 | 24.8 | 12.3 | 25.1 |
| Vision-R1-7B (Huang et al., 2025) | 36.7 | 43.7 | 47.4 | 57.3 | 38.7 | 33.7 | 29.2 | 24.5 | 52.3 | 29.4 | 10.6 |
| Qwen2.5-VL-32B-instruct (Bai et al., 2025) | 41.8 | 51.2 | 68.3 | 53.2 | 47.7 | 39.6 | 31.8 | 65.1 | 22.9 | 16.2 | 17.4 |
| Qwen2.5-VL-72B-instruct (Bai et al., 2025) | 43.7 | 53.5 | 36.2 | 63.9 | 61.3 | 49.8 | 33.0 | 29.9 | 37.8 | 29.9 | 35.2 |
| GLM-4.5V (Team et al., 2025) | 53.7 | 69.1 | **71.9** | **75.8** | 68.4 | 61.2 | 37.2 | 46.5 | 42.5 | 31.4 | 26.6 |
| Llama4-Maverick-17B-128E-FP8 (Meta, 2025) | **66.9** | **70.1** | 64.8 | 71.8 | **71.1** | **71.8** | **63.4** | 61.4 | 61.7 | **77.0** | **54.1** |
| *Open-source LMMs (single-image input)* | | | | | | | | | | | |
| LLaVA-v1.6-vicuna-7B (Liu et al., 2024a) | 20.7 | 22.6 | 21.4 | 8.4 | **32.4** | 26.3 | 18.7 | 29.5 | 2.8 | 23.0 | 18.4 |
| MiniCPM-v2.5 (Abdin et al., 2024) | 21.0 | 21.7 | 28.1 | 13.2 | 12.1 | 34.1 | 20.2 | 28.2 | 15.4 | 6.4 | **29.5** |
| LLaVA-onevision-7B (Li et al., 2024a) | 22.7 | 19.8 | **79.1** | 7.0 | 3.9 | 1.6 | 25.6 | 70.1 | 20.6 | 3.4 | 2.4 |
| LLaVA-v1.6-mistral-7B (Liu et al., 2024a) | 23.0 | 19.9 | 73.5 | 3.1 | 12.9 | 0.8 | **26.3** | **78.4** | 2.3 | 16.7 | 0.0 |
| LLaVA-v1.5-7B (Liu et al., 2023) | 23.7 | 23.6 | 23.5 | 19.4 | 12.5 | **38.4** | 23.8 | 33.6 | 17.3 | 14.2 | 28.5 |
| GLM4V-9B (GLM, 2024) | 23.9 | **25.6** | 19.4 | **31.7** | 31.6 | 18.8 | 22.2 | 10.3 | **33.2** | **26.0** | 20.8 |
| LLaVA-v1.6-vicuna-13B (Liu et al., 2024a) | **24.4** | 23.0 | 50.5 | 2.2 | 5.1 | **38.4** | 26.0 | 66.4 | 0.0 | 2.9 | 28.5 |
| *Open-Source LMMs (math-oriented)* | | | | | | | | | | | |
| USRA (Luo et al., 2025) | 27.4 | 28.9 | 31.6 | 30.2 | 32.0 | 22.9 | 26.0 | 38.3 | 30.1 | 21.5 | 12.3 |
| USRA-PS-RPO (Luo et al., 2025) | 22.8 | 23.8 | **55.1** | 14.5 | 6.3 | 25.5 | 21.8 | **44.4** | 26.2 | 1.5 | 11.1 |
| MM-Eureka-7B (Meng et al., 2025) | 37.9 | 50.9 | 36.2 | **62.1** | 52.7 | 50.1 | 24.0 | 21.1 | 22.4 | 27.4 | **25.6** |
| MM-Eureka-7B-CPGD (Liu et al., 2025) | **39.3** | **51.0** | 33.2 | 54.2 | **61.3** | **51.4** | 26.9 | 16.2 | 29.9 | 39.7 | 23.7 |
| MM-PRM-8B (Du et al., 2025) | 31.7 | 38.4 | 28.1 | 43.2 | 44.9 | 35.7 | 24.4 | 10.8 | **41.6** | 35.3 | 11.6 |
| MathCoderVL-2B (Wang et al., 2025a) | 24.5 | 25.9 | 19.4 | 23.3 | 42.6 | 15.2 | 23.3 | 16.6 | 13.6 | 50.5 | 14.5 |
| MathCoderVL-8B (Wang et al., 2025a) | 31.5 | 33.8 | 24.0 | 44.3 | 34.0 | 31.4 | **29.0** | 27.0 | 35.0 | 31.9 | 22.2 |
| PUMA7B (Zhuang et al., 2024) | 24.6 | 24.0 | 19.9 | 2.6 | 38.7 | 31.3 | 25.2 | 11.6 | 0.4 | **74.5** | 17.9 |
| VLM-R1-Math3B (Shen et al., 2025) | 27.4 | 29.3 | 32.1 | 30.8 | 32.0 | 23.1 | 25.2 | 39.0 | 31.3 | 19.1 | 9.2 |

Table 8: Performance comparison on VisioMath with results categorized based on image similarity.

| Models \ Image similarity | Avg | [0.16,0.68] | (0.68,0.87] | (0.87,0.96] | (0.96,1] |
|---|---|---|---|---|---|
| Human | 91.3 | 95.7 | 91.2 | 87.6 | 89.0 |
| Random | 25.6 | 23.6 | 24.4 | 27.8 | 27.1 |
| *Closed-source LMMs* | | | | | |
| GLM4V-plus (GLM, 2024) | 27.9 | 29.6 | 32.9 | 23.3 | 25.8 |
| QwenVL-plus (Bai et al., 2023) | 32.9 | 33.3 | 37.8 | 32.4 | 28.2 |
| QwenVL-max (Bai et al., 2023) | 44.1 | 47.3 | 50.2 | 41.3 | 37.6 |
| GPT-4o (OpenAI, 2024) | 45.9 | 53.8 | 50.9 | 40.0 | 39.1 |
| GPT-4.1 (OpenAI, 2025) | 52.6 | 65.8 | 56.4 | 42.9 | 45.1 |
| Gemini2-flash-thinking (DeepMind, 2025b) | 53.2 | 63.6 | 58.9 | 48.2 | 42.2 |
| Gemini2-flash (DeepMind, 2025a) | 55.5 | 66.7 | 59.8 | 49.3 | 46.2 |
| Doubao-1.5-Vision-pro (Team, 2025a) | 66.3 | 74.9 | 68.2 | 60.2 | 62.0 |
| Seed1.6-Thinking (ByteDance, 2024) | 72.3 | 82.4 | 74.2 | 66.2 | 66.4 |
| Gemini 2.5 Pro (Comanici et al., 2025) | **80.9** | **86.2** | **83.8** | **76.7** | **76.9** |
| *Open-source LMMs (multi-image input)* | | | | | |
| DeepSeekVL2-tiny (Wu et al., 2024b) | 23.5 | 23.3 | 24.0 | 24.4 | 22.4 |
| InternVL2.5-2B-MPO (Chen et al., 2024a) | 23.9 | 24.0 | 27.6 | 24.0 | 20.2 |
| InternVL2.5-2B (Chen et al., 2024a) | 24.6 | 24.2 | 28.9 | 22.7 | 22.7 |
| Llama3.2-11B-Vison (Dubey et al., 2024) | 25.3 | 23.3 | 27.8 | 26.4 | 23.6 |
| Idefics3-8B-llama (Laurençon et al., 2024) | 25.4 | 26.9 | 26.0 | 22.7 | 26.0 |
| Qwen2.5-VL-3B-instruct (Bai et al., 2025) | 25.4 | 26.7 | 27.6 | 24.4 | 22.9 |
| Phi3.5-vision (Abdin et al., 2024) | 25.7 | 23.6 | 28.7 | 27.8 | 22.9 |
| DeepSeekVL2-small (Wu et al., 2024b) | 26.6 | 30.7 | 29.6 | 24.9 | 21.3 |
| Mantis-8B-Idefics2 (Jiang et al., 2024) | 27.9 | 32.2 | 28.9 | 24.7 | 26.0 |
| InternVL2.5-4B (Chen et al., 2024a) | 28.2 | 28.9 | 31.8 | 27.3 | 24.7 |
| InternVL2.5-4B-MPO (Chen et al., 2024a) | 28.4 | 28.2 | 34.0 | 26.2 | 25.1 |
| MiniCPM-o-2.6 (Abdin et al., 2024) | 29.3 | 34.9 | 35.3 | 24.4 | 22.4 |
| R1-Onevison-7B (Yang et al., 2025) | 29.6 | 21.9 | 32.2 | 28.9 | 11.6 |
| MiniCPM-V-2.6 (Abdin et al., 2024) | 29.7 | 30.7 | 34.9 | 28.4 | 24.7 |
| InternVL2.5-8B (Chen et al., 2024a) | 29.9 | 32.4 | 31.8 | 29.6 | 26.0 |
| InternVL2.5-8B-MPO (Chen et al., 2024a) | 30.9 | 35.6 | 37.1 | 25.8 | 25.1 |
| QvQ-72B-Preview (Team, 2024b) | 30.9 | 37.3 | 38.0 | 25.3 | 23.1 |
| Qwen2-VL-72B-instruct (Wang et al., 2024c) | 31.7 | 35.5 | 37.8 | 26.0 | 27.1 |
| Qwen2.5-VL-7B-instruct (Bai et al., 2025) | 32.7 | 33.6 | 37.8 | 29.8 | 29.6 |
| Gemma3-27B (Team, 2025b) | 35.3 | 43.3 | 41.2 | 29.6 | 26.4 |
| Vision-R1-7B (Huang et al., 2025) | 36.7 | 46.7 | 38.9 | 30.4 | 30.9 |
| Qwen2.5-VL-32B-instruct (Bai et al., 2025) | 41.8 | 50.0 | 46.2 | 38.4 | 32.7 |
| Qwen2.5-VL-72B-instruct (Bai et al., 2025) | 43.7 | 47.1 | 50.8 | 38.0 | 38.7 |
| GLM-4.5V (Team et al., 2025) | 53.7 | **68.7** | 59.3 | 44.2 | 44.7 |
| Llama4-Maverick-17B-128E-Instruct-FP8 (Meta, 2025) | 66.9 | 63.6 | **70.0** | 65.8 | 68.2 |
| *Open-source LMMs (single-image input)* | | | | | |
| LLaVA-v1.6-vicuna-7B (Liu et al., 2024a) | 20.7 | 22.2 | 24.4 | 17.8 | 18.4 |
| MiniCPM-V-2.5 (Abdin et al., 2024) | 21.0 | 21.7 | 21.3 | 20.6 | 20.2 |
| LLaVA-onevision-7B (Li et al., 2024a) | 22.7 | 22.2 | 22.4 | 25.6 | 20.9 |
| LLaVA-v1.6-mistral-7B (Liu et al., 2024a) | 23.0 | 21.8 | **26.0** | 23.6 | 20.5 |
| LLaVA-v1.5-7B (Liu et al., 2023) | 23.7 | 23.3 | 25.3 | 24.9 | 21.1 |
| GLM4V-9B (GLM, 2024) | 23.9 | **26.7** | 23.5 | 23.3 | **22.0** |
| LLaVA-v1.6-vicuna-13B (Liu et al., 2024a) | **24.4** | 24.0 | **26.0** | **26.0** | 21.8 |
| *Open-source LMMs (math-oriented)* | | | | | |
| USRA-PS-RPO (Luo et al., 2025) | 22.8 | 26.7 | 22.0 | 24.0 | 18.7 |
| MathCoderVL-2B (Wang et al., 2025a) | 24.5 | 24.7 | 25.1 | 26.0 | 22.2 |
| PUMA7B (Zhuang et al., 2024) | 24.6 | 26.2 | 25.8 | 23.8 | 22.4 |
| USRA (Luo et al., 2025) | 27.4 | 26.0 | 31.4 | 26.3 | 26.1 |
| VLM-R1-Math3B (Shen et al., 2025) | 27.4 | 25.3 | 32.0 | 25.8 | 26.4 |
| MathCoderVL-8B (Wang et al., 2025a) | 31.5 | 33.8 | 37.1 | 28.4 | 26.7 |
| MM-PRM-8B (Du et al., 2025) | 31.7 | 37.6 | 37.1 | 26.9 | 25.1 |
| MM-Eureka-7B (Meng et al., 2025) | 37.9 | 45.6 | 44.0 | 29.1 | **33.1** |
| MM-Eureka-7B-CPGD (Liu et al., 2025) | **39.4** | **47.8** | **46.0** | **30.9** | 32.9 |

Table 9: This table presents the prompts used for process evaluation and answer generation by various LMMs in the VisioMath benchmark.

| Phase | Input | Prompt |
|---|---|---|
| **Answer Extraction** (GLM4-Flash) | Model's response | You are an AI assistant that helps me extract the answers to single-choice questions. You will be provided with an answer. Your task is to find the final option of the model. If the model's answer is meaningless, output Z. You should output a single uppercase letter, such as A, B, C, D (if they are valid options), or Z. **Example 1**: Answer: According to the question description and all related pictures, option A is the correct answer. Option A is a centrally symmetric figure because its four vertices are all symmetric, while the vertices of options B, C, and D are not symmetric. Output: A **Example 2**: Answer: A. Sphere B. Circle C. Disc D. Circle Output: Z **Example 3**: **Answer**: {model answer} **Output**: |
| **Answer Generation** (LMMs) | Question Diagrams | Please solve a single-choice math question. The last four pictures are respectively the pictures for options A, B, C, and D. Select the correct answer from A, B, C, and D based on the question description and all relevant pictures. |
| **Option Shuffling Generation** (LMMs) | Question Diagrams | Please solve a single-choice math question. The last four pictures are respectively the pictures for options B, C, Dand A. Select the correct answer from A, B, C, and D based on the question description and all relevant pictures. |
| **Image Caption Generation** (LMMs) | Question Diagrams | I have multiple images and a question that I want you to answer. I need you to strictly follow the format with three specific sections: SUMMARY, CAPTION and REASONING. To explain further: In SUMMARY, briefly explain what steps you'll take to solve the problem. In CAPTION, describe the contents of all the images, wrapping each image description inside tags like `<image1></image1>`, `<image2></image2>`, etc. In REASONING, outline a step-by-step thought process you would use to solve the problem based on the images. `<SUMMARY>` `[Summarize how you will approach the problem...]` `</SUMMARY>` `<CAPTION>` `<image1>...  </image1>` `<image2>...  </image2>` `</CAPTION>` `<REASONING>` `[Provide a chain-of-thought, logical explanation of the problem. This should outline step-by-step reasoning based on all the images.]` `</REASONING>` |
| **CoT Data Generation** (DeepSeek-V3.1) | Question Caption | Please solve this multiple-choice math question and answer in English. The last four images correspond to options A, B, C, and D respectively. Based on the question description and all relevant images, select the correct answer from A, B, C, and D. |

