# OpenReview forum: "VisioMath: Benchmarking Figure-based Mathematical Reasoning in LMMs"
_ICLR.cc/2026/Conference — ICLR 2026 Poster_

### Official Review · Reviewer_4GQo · 2025-10-15

**Soundness:** 2
**Presentation:** 2
**Contribution:** 2
**Rating:** 6
**Confidence:** 5

**Summary:**

The paper introduces VisioMath, a curated benchmark comprising 1,800 K–12 mathematics problems in which all answer options are diagrams that are highly visually similar. VisioMath aims to rigorously evaluate Large Multimodal Models (LMMs) on fine-grained, comparative mathematical reasoning tasks, particularly focusing on challenges where multiple image options differ only subtly. The authors present an in-depth evaluation of various state-of-the-art LMMs in both closed- and open-source families, reveal systematic weaknesses related to image–text misalignment, and propose alignment-oriented strategies that yield meaningful performance improvements. The benchmark, dataset construction pipelines, error analyses, and strategic interventions are all discussed in detail, with extensive experimental results and visualizations supporting the analysis.

**Strengths:**

1. VisioMath fills an under-explored niche by targeting figure-based multiple-choice mathematical reasoning where distractors are intentionally visually similar.
2. offers insightfully categorizes failures (image–text misalignment predominates) and connects these with systematic bottlenecks in LMM architectures. The controlled shuffling and ablation study setting is quite nice, offering insights to advance the multimodal reasoning domain
3. OK coverage of current vision or multimodal models with regards to evaluation and with perturbation setting to gauge the robustness of eval results

**Weaknesses:**

1. Limited Scope and Generalization: K12 and a quite narrow domain (geometry, algebra, etc.). The conclusions regarding LMMs’ comparative reasoning ability may not generalize to other STEM domains such as physics diagrams or scientific reasoning in general. If the authors want to target K12 for its pedagogical value maybe there should be a clear case made for that in analysis etc.

2. Comparison with related work: more multimodal benchmark that explores vision-dependency of reasoning can be compared and discussed like SeePhys, MMSciBench, VisAidedMath etc. that goes beyond the current limited scope to position this work better in the landscape of multimodal reasoning, the authors could further strengthen this work by comparing the vision-based approach in these related works. The current first para of related work on LMM is really not that necessary and cannot possibly cover all LMMs due to the nature of that field's progress.

3. Human baseline? Are these similarities only confusing for models or for humans as well? I also am a bit confused as to the necessity of classifying open-source LMMs into single vs. multi-image input? If they cannot process multiple images then they cannot process all the options correctly right?

PS the current abstract entry on OpenReview has a entry typo "Large" missing an "L"

**Questions:**

1. Many of the tables are too dense and hard to read (eye-straining already) or grasp key points e.g. by color highlights or optionally move something into the appendix.

2. Wording can be improved, "question stem length" is confusing, "visual-option" should perhaps be "image-option" because otherwise I can easily understand it as "visual-optional" so overall the presentation needs to be improved, a lot of the wording are highly confusing and will benefit significantly from more polishing in phrasing.

In general I think this design is really nice for testing vision-based reasoning capability for LMMs and the shuffing/strategy experiments are quite nice as analysis. Although the overall presentation could be significantly improve to aid readability of this manuscript I think this paper has merited a OK contribution to this field.

---

> ### Author Response · Authors · 2025-11-24
> **Response to reviewer 4GQo**
>
> Thank you for your valuable feedback.
>
> Here we address the points you mentioned in the weaknesses part.
>
> 1. We thank the reviewer for this insightful comment. While VisioMath currently focuses on K–12 mathematics, this choice is deliberate due to both pedagogical significance and the unique challenges posed by visually similar diagrams in mathematics education. Fine-grained comparative reasoning is central to K–12 learning, where students must distinguish nearly identical diagrams to identify correct solutions. This makes it an ideal testbed for evaluating the visual-textual grounding capabilities of LMMs. Moreover, insights gained from this domain are likely transferable to other STEM areas that involve diagrammatic reasoning, such as physics or chemistry, because the core challenge, aligning multiple visually similar inputs with textual cues, remains analogous. **We have clarified this motivation in the revised manuscript (line 91-95), highlighting both the methodological relevance and the educational impact.**
> 2. We thank the reviewer for highlighting the importance of situating VisioMath within the broader landscape of multimodal reasoning benchmarks. Benchmarks such as SeePhys[1], MMSciBench[2], and VisAidedMath[3] indeed investigate reasoning tasks that require visual grounding, and we agree that they reflect an important direction in multimodal evaluation. However, these benchmarks mostly target single-image settings, differ from our setting in a fundamental way: their problems do not require _fine-grained, multi-image comparative reasoning over visually similar diagrams_, which is the central capability VisioMath is designed to isolate and measure. **In the updated manuscript, we follow the suggestion and remove the first section of the Related Work, and add the additional references, along with corresponding discussion (line 440-447), into the “Multimodal Understanding Benchmarks” subsection.**
>
> References:
>
> [1] Xiang et al., SeePhys: Does Seeing Help Thinking? -- Benchmarking Vision-Based Physics Reasoning. NeurIPS 2025.
>
> [2] Ye et al., MMSciBench: Benchmarking Language Models on Chinese Multimodal Scientific Problems. ACL 2025.
>
> [3] Ma et al., VisAidMath: Benchmarking Visual-Aided Mathematical Reasoning. NeurIPS 2025 Workshop.
>
> 3. (1) We appreciate the reviewer for raising this important point. We conducted a preliminary human study on a randomized subset of 180 problems and report the results in the table below. First, human performance moderately decreases as visual similarity among candidate diagrams increases, indicating that higher similarity introduces additional perceptual challenge. Second, beyond a certain similarity threshold, accuracy plateaus, suggesting that errors at this stage are driven more by conceptual understanding than by perceptual similarity. In contrast, as shown in the left of Figure 10 in the appendix, LMMs often fail on perceptually trivial distinctions, which humans rarely confuse. These findings indicate that enhancing visual–textual alignment in LMMs could substantially reduce such errors. **We are currently running a full-scale human evaluation over the entire benchmark and will include the complete results in the revised version before the rebuttal deadline.**
>
>
> |  | **Avg** | **[0.16, 0.68]** | **(0.68, 0.87]** | **(0.87, 0.96]** | **(0.96, 1]** |
> |--| --- | --- | --- | --- | --- |
> | Human | 88.3 | 95.6 | 86.7 | 82.2 | 88.8 |
>
> (2) We categorize open-source LMMs as single- versus multi-image to highlight fundamental architectural differences. Single-image models (e.g., LLaVA-v1.6) cannot natively process multiple inputs simultaneously. To enable inference, we presented them with a consolidated “stitched-image” layout (Strategy 1), allowing access to all candidate options. Despite this, their performance remains substantially lower than multi-image models, indicating that limitations in cross-image reasoning rather than lack of visual information drive the observed deficiencies.
>
> Then here are the answers to the Questions part.
>
> 1. We appreciate the reviewer’s feedback regarding table density and readability. To improve clarity, we have revised the tables to highlight key points. Specifically, several models in Tables 2 and 3 have been moved to the appendix, retaining only the high-performing models in the main text.
> 2. We thank the reviewer for the constructive feedback on wording and presentation. In response, we replace “question stem length” with "question text length", and replace “visual-option” with “image-option” to avoid potential misinterpretation. More generally, we systematically refined phrasing throughout the paper to improve readability and reduce confusion.

---

> ### Comment · Reviewer_4GQo · 2025-11-24
>
> Thank you for the response, I think the additional human experiment, improved presentation and related work has significantly improved the quality and value of this work. And I think K-12 design is acceptable given the intention of the authors on its pedagogical background, which is also (one of) the key reasons why I find human study to be particularly important since pedagogy at the end needs to target human students (often young students) so it might be a interesting follow-up to see how well K-12 students of various age/skill level and with various educational background (or personal conditions in general) perform on this benchmark, or how well the LM-performance is equivalent to students of age=X. The addition of more related work gives us a better sense to position the contribution of this work in the LMM-eval landscape.
>
> Nevertheless, I find the authors' response and improvements to be satisfying, and I hope to see more thorough human study (e.g. more/diverse candidates, characterization of human subjects, test on whole benchmarks, cross-comparison with pedagogical/cognitive analysis) and related work in its camera-ready version.
>
> Overall, I think this work ought to be presented at ICLR and recommend acceptance to the AC. As such, I am raising my score from 6 to 8.
>
> (PS I still don't really think the single-image models necessitate presentation in the main text as they are presumably inferior anyways, so that result is not surprising, maybe putting into appendix can save more space for more important contents)

---

> ### Author Response · Authors · 2025-11-24
>
> Thank you for your feedback and updated scores. We appreciate the recognition of our added human experiment, improved presentation, and expanded related work. We agree that broader human-subject studies across diverse ages and learner profiles are an important direction. Regarding the single-image models, we will move them to the appendix for more essential content.

---

> ### Author Response · Authors · 2025-12-02
>
> Here we provide our human baseline results.
>
> 1. **Further response to Weakness 3.**
> We appreciate the reviewer for raising this important point. To establish a rigorous upper bound for model performance, we have now completed a **full-scale human evaluation** on the entire benchmark.  The complete results are reported in the table below, as well as in **Tables 2, 3, 7, and 8** of the revised manuscript.
>
>     | | **Avg** | **[0.16, 0.68]** | **(0.68, 0.87]** | **(0.87, 0.96]** | **(0.96, 1]** |
>     | --- | --- | --- | --- | --- | --- |
>     | Human | 91.3 | 95.7 | 91.2 | 87.6 | 89.0 |
>
>
> **These full-scale results strongly support our initial conclusions.** First, human performance moderately decreases as visual similarity among candidate diagrams increases, confirming that higher similarity introduces additional perceptual challenges. Second, beyond a certain similarity threshold, accuracy plateaus, suggesting that errors at this stage are driven more by conceptual understanding than by perceptual similarity. This suggests that while high similarity increases perceptual load, humans can still reliably distinguish fine-grained differences through careful observation. In contrast, as shown in the left panel of Figure 10 (Appendix), LMMs often fail on perceptually trivial distinctions that humans rarely confuse. These findings reinforce our argument that enhancing visual–textual alignment in LMMs is critical to reducing such errors.

---

### Official Review · Reviewer_bDac · 2025-10-28

**Soundness:** 3
**Presentation:** 2
**Contribution:** 2
**Rating:** 4
**Confidence:** 4

**Summary:**

This paper introduces VisioMath, a new benchmark designed to address the evaluation gap in the fine-grained comparative reasoning abilities of current Large Multimodal Models when handling multiple, visually similar options. It presents VisioMath, a benchmark comprising 1,800 high-quality K-12 mathematics problems. Its unique characteristic is that all candidate answers are visually highly similar diagrams, compelling models to perform fine-grained visual discrimination and comparative reasoning, rather than simple pattern matching. Through extensive experiments on a series of SOTA models, the paper reveals a common deficiency in current models on this task, with the root cause being primarily image-text misalignment. Finally, the paper explores and validates three effective alignment strategies that significantly improve model performance. Its core contribution lies in providing a new, challenging evaluation tool, while also offering deep insights and practical directions for enhancing the multimodal alignment capabilities of LMMs.

**Strengths:**

- The paper identifies and addresses a critical problem in the field of LMM evaluation by focusing on visually similar diagrammatic options.

- Through the quantification of visual similarity and an innovative "option shuffling" experiment, it uncovers that the core weakness of current LMMs lies in precise image-text alignment.

- It proposes three concrete and viable performance enhancement strategies ranging from training-free methods to lightweight fine-tuning and experimentally demonstrates their effectiveness, offering the community a clear roadmap to tackle this issue.

**Weaknesses:**

- The definition of visual similarity relies entirely on a single model (Qwen multimodal-embedding-v1). Different models' visual encoders may have varying interpretations of "similarity." A brief discussion or experiment demonstrating the correlation or discrepancy with other metrics (such as CLIP embeddings or DINO scores), or one that further justifies this choice, would make this core metric more robust.

- The fine-tuning experiment for Strategy 3 was conducted on only a single model (Qwen2.5-VL-3B). Applying this method to another model family with a different architecture and observing similar results would provide stronger evidence for the generality of this conclusion.

- The analysis of "Vision Recognition Errors" lacks depth. At 34%, this error type represents a significant proportion. Are these errors due to the model's inability to recognize basic geometric shapes (e.g., confusing a triangle with a pyramid), or its failure to understand more abstract symbols (such as the monotonicity of a function's graph)? A more fine-grained error classification would offer greater insight into the limitations of current visual encoders in diagram understanding.

- The dataset is sourced exclusively from Chinese exams. Although mathematics is a universal language, the phrasing of the problems and the style of the diagrams may still carry certain cultural or curricular imprints.

- In the "option shuffling" experiment, when faced with misaligned labels, do the models tend to guess randomly, or do they systematically follow the new (but incorrect) positional cues (e.g., always choosing the image corresponding to the new position of label 'A')? This could provide deeper insight into the nature of the heuristic rules the models rely on.

**Questions:**

same as weakness

---

> ### Author Response · Authors · 2025-11-24
> **Response to reviewer bDac**
>
> Thank you for taking the time to review our paper.
>
> Here we address the points you mentioned in the weaknesses part.
>
> 1. We thank the reviewer for raising this point. Our choice of Qwen multimodal-embedding-v1 is primarily justified by its training regime: unlike CLIP, which is optimized for natural image–text pairs, and BLIP, which focuses on general QA tasks, Qwen has been trained on diverse multimodal datasets that include diagrammatic reasoning and text–diagram QA pairs. This background makes its embeddings particularly suitable for capturing subtle, semantics-oriented similarities in mathematical diagrams.
>
>     To further validate this choice, we conducted a Nearest Neighbor Retrieval experiment using different models including two additional candidates (BLIP and CLIP) to compute visual similarity. **We show qualitative examples in Figure 7 of the appendix in the revised manuscript.** Across these examples, we can see that Qwen’s embeddings consistently retrieved diagrams that better align with human judgment, demonstrating its superior semantic grounding for the types of fine-grained visual comparisons central to VisioMath.
>
> 2. We thank the reviewer for highlighting the question of generalizability for Strategy 3. To address this, we extended our experiments beyond the original Qwen2.5-VL-3B model. Specifically, we evaluated Strategy 3 on larger models within the Qwen2.5-VL family (7B and 72B) as well as on a model with a different architecture, InternVL2.5-2B. The results, summarized in the table below, show consistent and substantial accuracy improvements across all models, demonstrating that the benefits of Strategy 3 are both scalable and architecture-agnostic. **We also added these results in the revised version in Table 4.**
>
>     | **Model** | **Original** | **Strategy 3** |
>     | --- | --- | --- |
>     | Qwen2.5-VL-7B-instruct | 32.7 | 43.3 (**+10.6**) |
>     | Qwen2.5-VL-72B-instruct | 43.7 | 51.4 (**+7.7**) |
>     | InternVL2.5-2B | 24.6 | 32.2 (**+7.6**) |
>
>
> 3. **We further conducted a detailed analysis of vision recognition errors,  with their distributions and representative examples provided in Appendix D.1 in the revised manuscript.** We classify these errors into four categories:
>
>     **(1) Fine-Grained Geometric Errors (18%)** – where models cannot discriminate subtle quantitative differences between visually similar options;
>
>     **(2) Spatial Topology & Transformation Errors (24%)** – arising from difficulties in understanding 2D-to-3D mappings (e.g., folding nets) or element connectivity;
>
>     **(3) Spatial Positional Relation Errors (22%)** – reflecting challenges in judging relative positions within coordinate systems or among diagram components;
>
>     **(4) Abstract Global Pattern Recognition Errors (36%)** – the most frequent type, where models fail to integrate local features into a coherent global geometric pattern or structural layout.
>
> 4. We acknowledge that our primary dataset is sourced from Chinese exams; however, mathematics as a discipline is largely universal, and the visual reasoning patterns they instantiate—such as comparing geometric configurations, identifying topological differences, and mapping text to diagrams—are not tied to culture-specific content.
>
>     To further assess potential sourcing bias, we additionally collected a supplementary set of 292 additional problems from international competitions (e.g., AMC, Kangaroo Math). These problems were processed through the same rigorous curation pipeline described in Section 2.1, ensuring uniform quality, consistent LaTeX rendering, and controlled visual similarity across all items. We show several representative models on this dataset in the table below. Importantly, the results show a consistent pattern with our main findings, in particular, accuracy degradation with increasing inter-image similarity remain unchanged.
>
>     | **Model** | **Overall** | **Question stem w/o images** | **Question stem with images** | **Q1** | **Q2** | **Q3** | **Q4** |
>     | --- | --- | --- | --- | --- | --- | --- | --- |
>     | **GPT4o** | 30.8 | 36.4 | 28.4 | 47.9 | 27.4 | 26.0 | 21.9 |
>     | **QwenVL2.5_3b** | 24.7 | 25.4 | 22.7 | 28.7 | 26.0 | 23.2 | 20.5 |
>     | **GLM4.5v** | 32.5 | 52.3 | 24.0 | 39.7 | 32.9 | 27.4 | 30.1 |
>     | **Seed1.6** | 44.5 | 59.1 | 38.2 | 60.3 | 41.1 | 37.0 | 39.7 |
>
>
> 5. To investigate this, we analyzed the cases in which Gemini 2.5 Pro originally answered correctly but failed after option shuffling. We found that 76.8% of these failures arose from the model’s reliance on an implicit positional prior—specifically, selecting the image that would have been correct under the original A–B ordering. This indicates a systematic positional bias in how the model interprets image order. These results substantiate that “image–text misalignment” is a fundamental limitation: the model frequently defaults to shallow positional heuristics rather than grounding its reasoning in the semantic content.

---

### Official Review · Reviewer_9Krr · 2025-10-31

**Soundness:** 3
**Presentation:** 3
**Contribution:** 3
**Rating:** 6
**Confidence:** 4

**Summary:**

This paper introduces VisioMath, a new benchmark to evaluate the fine-grained, multi-image comparative reasoning of Large Multimodal Models (LMMs). The authors identify a critical gap in existing benchmarks: LMMs are rarely tested on problems common in K-12 mathematics, where all answer options are visually similar diagrams (e.g., graphs, geometric figures).

The VisioMath benchmark consists of 1,800 high-quality, curated K-12 math problems sourced from real-world exams. Its defining feature is that all answer options are diagrams, and the dataset includes a metric to quantify the (often high) visual similarity between them.

Through a comprehensive evaluation of state-of-the-art LMMs (e.g., GPT-4.1, Gemini 2.5 Pro, Qwen2.5-VL), the paper demonstrates:

* A consistent decline in accuracy as the visual similarity between answer options increases.
* The dominant failure mode (36% of errors) is image-text misalignment. Models rely on shallow positional heuristics (e.g., assuming the first image corresponds to option A) instead of true semantic grounding.
* This reliance on heuristics is proven by a controlled option shuffling experiment, where permuting the text labels (e.g., "A, B, C, D" → "B, C, D, A") while keeping images in the same order caused a significant drop in accuracy (e.g., -8.7% for Gemini 2.5 Pro).

Finally, the authors propose and validate three strategies to mitigate this: two training-free (consolidating images into a single layout and adding explicit visual-textual anchors) and one training-based (alignment-oriented CoT finetuning), which achieved a +12.6% accuracy gain on Qwen2.5-VL-3B.

**Strengths:**

* Originality and Significance: The paper identifies and rigorously tests a novel, practical, and highly significant problem. The task of "figure-based option" reasoning, especially with high-similarity distractors, is a ubiquitous real-world scenario (especially in STEM education) that has been almost entirely overlooked by existing benchmarks.
* Benchmark Quality: The VisioMath benchmark is meticulously constructed and serves as an excellent diagnostic tool, not just a leaderboard. The curation from real exams ensures representativeness. Key quality steps include enforcing a one-image-per-option rule, balancing the answer distribution to prevent bias, and, crucially, quantifying the visual similarity of options to enable controlled analysis.
* Analytical Depth: The paper's primary strength is its deep analysis, which goes far beyond reporting accuracy. The option shuffling experiment (Fig 4b) is a simple but brilliant controlled study that provides definitive proof of the model's reliance on positional heuristics. This insight is arguably more valuable than the benchmark itself. The error categorization (Fig 4a), which pinpoints "image-text misalignment" as the dominant failure, is also a key contribution.
* Clarity: The paper is very well-written, with clear and effective figures. Figure 1 instantly communicates the problem, Figure 2 shows the careful data pipeline, and Figure 4 clearly presents the core analytical findings.

**Weaknesses:**

* Apparent Contradiction in Concatenation Strategy: There is a confusing contradiction in the results. The baseline evaluation (Table 2) shows that single-image LMMs, which use a "composite image concatenation strategy," perform at random-guess levels (e.g., LLaVA-v1.6 at 24.4%). However, "Strategy 1," which also uses a "consolidated single image layout" (concatenation), *improves* the performance of multi-image LMMs (e.g., +6.4% for Seed1.6-Thinking). The paper does not adequately explain why concatenation fails so catastrophically for one set of models but serves as an effective enhancement for another.
* Limited Scope of Finetuning (Strategy 3): The finetuning result (Strategy 3) is very promising, yielding a substantial +12.6% gain. However, this experiment was only conducted on a single, small model (Qwen2.5-VL-3B). This makes it difficult to assess if this is a generally effective strategy or if the gains would be as pronounced on larger, more capable models (e.g., 72B or closed-source models).
* Benchmark Sourcing: The dataset is sourced exclusively from Chinese high school and college entrance examinations. While the authors translate the problems, this single-source origin may limit the benchmark's global representativity, as it may contain cultural or pedagogical biases specific to that education system.
* Similarity Metric Rationale: The visual similarity for a question, Sim(Q), is defined as the *minimum* pairwise cosine similarity between options. This choice is not fully justified and could be fragile. For example, a question with three identical options and one complete outlier would receive a very low similarity score, even though it would be a very "confusing" set for a model. An average or median pairwise similarity might be a more robust measure of overall inter-option confusability.

**Questions:**

1. Could you please clarify the discrepancy between the failure of single-image LMMs using concatenation (Table 2) and the success of Strategy 1 (also concatenation) for multi-image LMMs (Fig 4b)? Is this difference due to (a) the models' pre-training (single-image vs. interleaved multi-image), (b) the specific concatenation method used (e.g., zero-padding), or (c) the inherent architectural differences in how these models process a composite image?

2. The +12.6% gain from your alignment-oriented CoT data is very impressive. To demonstrate the scalability of this approach, have you attempted to finetune other models in the Qwen2.5-VL family (e.g., the 7B or 72B versions) to see if this substantial gain is consistent across model scales?

3. What was the rationale for choosing the *minimum* pairwise similarity as your Sim(Q) metric? Have you analyzed how the results would change if you used the *average* or *median* pairwise similarity, which might be more robust to a single outlier option?

4. The gains from Strategy 2 (explicit anchors) vary wildly, from +9.8% for QwenVL-plus to only +0.9% for Gemini 2.5 Pro. What is your hypothesis for this large variance? Does it suggest that more advanced models like Gemini 2.5 Pro are already attempting this alignment (and thus don't benefit much from the explicit hint), whereas weaker models are not?

---

> ### Author Response · Authors · 2025-11-24
> **Response to reviewer 9Krr(1/2)**
>
> Thank you for your thoughtful comments and suggestions.
>
> Here we address the points you mentioned in the weaknesses part.
>
> 1. We thank the reviewer for highlighting the seemingly contradictory effects of image concatenation. Although both the Single-Image baseline and Strategy 1 employ composite-image layouts, the divergent outcomes originate from fundamental architectural and pre-training differences between Single-Image and Multi-Image LMMs, as well as the distinct bottlenecks each class of models faces.
>
>     **Single-Image LMMs:**
>     For models such as LLaVA-v1.6, stitching is merely a workaround to force all options into a single input. Their poor performance does not result from concatenation, but from an inherent limitation: these models are trained on isolated image–text pairs and therefore lack cross-image relational reasoning. When presented with a composite image, they fail to model relationships between them. Thus, concatenation exposes a missing capability rather than causing degradation.
>
>     **Multi-Image LMMs:**
>     In contrast, multi-image models are pre-trained on interleaved multimodal data and already possess latent comparative-reasoning ability. Their main difficulty in the standard multi-image setting is image–text misalignment. For these models, the same stitching strategy helps by consolidating visual information and improving grounding. Hence, concatenation acts as an _alignment aid_, leading to the observed performance gains.
>
>    In short, concatenation fails for Single-Image LMMs because they lack relational reasoning, but helps Multi-Image LMMs because it reduces cross-input alignment overhead.
>
> 2. We appreciate the reviewer’s concern regarding the generalizability of Strategy 3 beyond a single small model. In response, we conducted additional experiments on both larger models within the Qwen2.5-VL family (7B and 72B) as well as on a different architecture, InternVL2.5-2B. The results shown in below table consistently demonstrate substantial and scalable improvements. **We also added these results in the revised version in Table 4.**
>
>     | **Model** | **Original** | **Strategy 3** |
>     | --- | --- | --- |
>     | Qwen2.5-VL-7B-instruct | 32.7 | 43.3 (**+10.6**) |
>     | Qwen2.5-VL-72B-instruct | 43.7 | 51.4 (**+7.7**) |
>     | InternVL2.5-2B | 24.6 | 32.2 (**+7.6**) |
>
>
> 3. We appreciate the reviewer’s concern regarding the benchmark’s source distribution. While the problems originate from Chinese K–12 examinations, we note that the visual reasoning patterns they instantiate—such as comparing geometric configurations, identifying topological differences, and mapping text to diagrams—are not tied to culture-specific content. These tasks reflect universally taught mathematical concepts and diagrammatic conventions.
>
>     To further assess potential sourcing bias, we additionally collected a supplementary set of 292 additional problems from international competitions (e.g., AMC, Kangaroo Math). These problems were processed through the same rigorous curation pipeline described in Section 2.1, ensuring uniform quality, consistent LaTeX rendering, and controlled visual similarity across all items. We show several representative models on this dataset in the table below. Importantly, evaluation on this internationally diverse subset exhibits performance patterns that are highly consistent with those observed on the original dataset. In particular, accuracy degradation with increasing inter-image similarity remain unchanged.
>
>     | **Model** | **Overall** | **Question stem w/o images** | **Question stem with images** | **Q1** | **Q2** | **Q3** | **Q4** |
>     | --- | --- | --- | --- | --- | --- | --- | --- |
>     | **GPT-4o** | 30.8 | 36.4 | 28.4 | 47.9 | 27.4 | 26.0 | 21.9 |
>     | **QwenVL2.5_3b** | 24.7 | 25.4 | 22.7 | 28.7 | 26.0 | 23.2 | 20.5 |
>     | **GLM4.5v** | 32.5 | 52.3 | 24.0 | 39.7 | 32.9 | 27.4 | 30.1 |
>     | **Seed1.6** | 44.5 | 59.1 | 38.2 | 60.3 | 41.1 | 37.0 | 39.7 |
>
>
> 4. We appreciate the reviewer’s concern regarding the rationale for defining visual similarity using the minimum pairwise cosine similarity. Our choice is motivated by the need to capture the  _**strictest comparative difficulty**_ of a problem: the minimum similarity ensures that every diagram pair in a question is at least as similar as this threshold, thereby reflecting the lower bound of inter-option confusability.
>
>    To address the reviewer’s concern about potential fragility, we conducted an additional analysis using three alternative metrics—minimum, mean, and median pairwise similarity—across all 1,800 problems. **As shown in Appendix E (Figure 6)**, the distributions of these three metrics exhibit highly aligned quartile boundaries, indicating that the dataset’s grouping under different similarity definitions is structurally consistent rather than sensitive to a single metric.

---

> ### Author Response · Authors · 2025-11-24
> **Response to reviewer 9Krr(2/2)**
>
> Furthermore, we evaluated multiple models under quartile splits generated by each similarity measure. Across all settings, the central conclusion remains unchanged: model accuracy consistently decreases as similarity increases, regardless of whether similarity is defined by minimum, mean, or median pairwise scores. This robustness demonstrates that our key empirical findings do not depend on a particular choice of metric.
>
> We agree that certain edge cases (e.g., several nearly identical options plus one outlier) can occur. However, from a statistical perspective, such outliers do not meaningfully affect the overall characterization of the dataset.
>
>
>
> Then here are the answers to the Questions part.
>
> 1. Concatenation fails for single-image LMMs because it exposes a missing ability, multi-image relational reasoning, whereas it benefits multi-image LMMs by mitigating cross-input alignment overhead. Please refer to our response to Weakness 1 for more details.
> 2. we conducted additional experiments on larger models within the Qwen2.5-VL family (7B and 72B) as well as on a different architecture, InternVL2.5-2B. Please refer to our response to Weakness 2.
> 3. We use the minimum pairwise similarity because it ensures that all diagram pairs in a problem are at least as similar as this value, directly capturing the strictest comparative challenge. We also tested mean (average) and median similarities across all 1,800 problems; the overall trends and quartile divisions were similar. Please refer to our response to Weakness 4.
> 4. We thank the reviewer for this insightful observation. As noted, models such as QwenVL-plus exhibit a relatively low baseline accuracy (32.9%), reflecting difficulties in establishing reliable visual–textual correspondences across multiple similar images. For these models, Strategy 2’s explicit anchors effectively bridge this perceptual gap, yielding a substantial gain (+9.8%). In contrast, state-of-the-art models like Gemini 2.5 Pro already achieve a high baseline accuracy (80.9%), suggesting the presence of strong intrinsic spatial attention mechanisms that enable implicit alignment. Consequently, the additional benefit of explicit anchors is minimal (+0.9%). Overall, these results indicate that while Strategy 2 is most impactful for weaker models, it remains a useful and complementary tool for enhancing multi-image reasoning across model strengths.

---

### Official Review · Reviewer_ABAJ · 2025-10-31

**Soundness:** 3
**Presentation:** 3
**Contribution:** 3
**Rating:** 6
**Confidence:** 4

**Summary:**

This paper, *“VisioMath: Benchmarking Figure-Based Mathematical Reasoning in LMMs,”* presents a new benchmark designed to evaluate **fine-grained visual reasoning in mathematical contexts**, particularly when **multiple highly similar diagrams** are presented as answer choices.

The authors construct **VisioMath**, a dataset of **1,800 high-quality K–12 math problems** collected from real entrance exams (2002–2023). Each problem provides **four visually similar diagrammatic options (A–D)**, with some including **images in the question stem**. Such *diagram-based answer formats* are indeed rare among existing math benchmarks, making this a **valuable and innovative dataset contribution**.

The benchmark evaluates a wide range of **open- and closed-source large multimodal models (LMMs)**, and key findings include:

- **Multi-image reasoning remains a major weakness** — accuracy often drops to near chance level when multiple diagrams must be compared.
- **High visual similarity sharply reduces performance** — all models degrade significantly as image similarity increases, indicating weak fine-grained perception.
- **Positional bias exists** — models show uneven accuracy across option positions, especially when the question stem includes images.
- **Image–text misalignment dominates failure cases** — models frequently rely on superficial spatial cues rather than grounded reasoning.

To mitigate these issues, the authors explore **three alignment-oriented strategies**:
(1) merging all visuals into a **consolidated single layout**,
(2) overlaying **explicit textual anchors** on images, and
(3) fine-tuning with multi-image chain-of-thought (CoT) samples.
These approaches lead to **notable performance gains**, demonstrating that targeted alignment interventions can substantially improve multimodal reasoning in visually complex mathematical tasks.

**Strengths:**

1. **Novel Benchmark Design for Visual Reasoning in Math**
   The benchmark uniquely incorporates **both image-based question stems and diagrammatic answer options**, a setting that is rarely seen in prior math benchmarks.
   This integration of text and diagrams offers a more realistic and cognitively demanding setup, and I believe such *interleaved visual–textual formats* represent an important future direction for multimodal reasoning benchmarks.

2. **High-Quality, Real-World Data Source**
   The dataset is constructed from **authentic K–12 entrance exam problems**, ensuring **pedagogical validity**, **realistic problem difficulty**, and alignment with real-world educational reasoning tasks.
   This grounding in real exam materials enhances the dataset’s credibility and practical relevance.

3. **Effective Improvement Strategies**
   The paper proposes **three lightweight yet effective alignment-oriented interventions** — layout consolidation, explicit textual anchors, and multi-image CoT fine-tuning.
   These strategies are simple to implement but lead to **clear and interpretable performance gains**, demonstrating practical value for improving multimodal alignment in large models.

**Weaknesses:**

1. **Restricted Evaluation Format (Multiple-Choice Only)**
   Every question in VisioMath follows a **four-choice multiple-choice format**, which simplifies the reasoning process and limits evaluation to **discrete answer selection**.
   This design makes it difficult to assess open-ended or step-by-step reasoning ability, which is essential for understanding the full reasoning depth of large multimodal models.

2. **Lack of Failure Case Analysis Beyond Accuracy**
   As a benchmark paper, the analysis would be more insightful if it included **qualitative examples of failure cases**.
   For instance, showing what a model’s **chain-of-thought output looks like when cross-image attention fails** would help readers better understand the specific weaknesses and reasoning breakdowns of current LMMs.

**Questions:**

Overall, this is a very solid and well-executed paper.
I do have one question: when evaluating on the benchmark, why didn’t the authors include specialized math-oriented multimodal models for comparison?

---

> ### Author Response · Authors · 2025-11-24
> **Response to reviewer ABAJ**
>
> Thank you for your valuable feedback.
>
> Here we address the points you mentioned in the weaknesses part.
>
> 1. The use of multiple-choice format is an intentional design choice aligned with the core objective of VisioMath: isolating and evaluating _fine-grained comparative reasoning across visually similar diagrams_. This capability is difficult to assess in open-ended formats, where models can often generate correct answers by relying on text-only shortcuts, thereby bypassing the cross-image grounding that our benchmark is designed to test. Moreover, reliable and standardized automatic scoring for open-ended reasoning remains an open research problem. Developing such evaluation protocols is non-trivial, as free-form solutions introduce ambiguity in both representation and correctness criteria. We fully agree that evaluating open-ended or step-by-step reasoning is valuable, and consider open-ended extensions as a promising direction.
> 2. We agree that qualitative failure cases can deepen the analysis. We have provided detailed examples across four error types in Figure 8 and 9, Appendix D of the submission. In addition, we further expanded this section in the revised version (Appendix D.1) with a finer-grained analysis of Vision Recognition Errors to better illustrate how cross-image grounding fails in current LMMs.
>
> Then here are the answers to the Questions part.
>
> 1. We appreciate the reviewer’s insightful suggestion. In response, we have substantially broadened our experimental evaluation to include nine state-of-the-art, math-oriented multimodal models: VLM-R1-Math-3B [1], MathCoder-VL (2B & 8B) [2], Math-PUMA-7B [6], URSA (8B & PS-RPO-8B) [7], MM-Eureka (7B & 7B-CPGD) [3,4], and MM-PRM-8B [5]. The results are shown below.
>
> Our expanded analysis reveals a consistent pattern: although these specialized models achieve strong performance on conventional single-image mathematical reasoning tasks, their accuracy drops markedly under the fine-grained multi-image discrimination required by VisioMath. This empirical finding supports that domain-specific mathematical pretraining alone does not resolve the core challenge of comparative diagram understanding. The observed performance gap further underscores the need for improved multiple image–text grounding.
>
> **In the revised manuscript, we reported the complete results in Table 7 and 8, while Table 2 and 3 present the three highest-performing models for clarity.**
>
> References:
>
> [1] Shen et al., VLM-R1: A Stable and Generalizable R1-Style Large Vision-Language Model, arXiv 2025.
>
> [2] Wang et al., MathCoder-VL: Bridging Vision and Code for Enhanced Multimodal Mathematical Reasoning, ACL 2025.
>
> [3] Meng et al., MM-Eureka: Exploring the Frontiers of Multimodal Reasoning, arXiv 2025.
>
> [4] Du et al., CPGD: Toward Stable Rule-based Reinforcement Learning, arXiv 2025.
>
> [5] Liu et al., MM-PRM: Enhancing Multimodal Mathematical Reasoning, arXiv 2025.
>
> [6] Zhuang et al., Math-PUMA: Progressive Upward Multimodal Alignment, AAAI 2025.
>
> [7] Luo et al., URSA: Understanding and Verifying Chain-of-thought Reasoning, NeurIPS 2025.
>
>
>
>
>
> | **Models \ Image similarity** | **Avg** | **[0.16,0.68]** | **(0.68,0.87]** | **(0.87,0.96]** | **(0.96,1]** |
> | --- | --- | --- | --- | --- | --- |
> | USRA-PS-RPO | 22.8 | 26.7 | 22.0 | 24.0 | 18.7 |
> | MathCoderVL-2B | 24.5 | 24.7 | 25.1 | 26.0 | 22.2 |
> | PUMA7B | 24.6 | 26.2 | 25.8 | 23.8 | 22.4 |
> | USRA | 27.4 | 26.0 | 31.4 | 26.3 | 26.1 |
> | VLM-R1-Math3B | 27.4 | 25.3 | 32.0 | 25.8 | 26.4 |
> | MathCoderVL-8B | 31.5 | 33.8 | 37.1 | 28.4 | 26.7 |
> | MM-PRM-8B | 31.7 | 37.6 | 37.1 | 26.9 | 25.1 |
> | MM-Eureka-7B | 37.9 | 45.6 | 44.0 | 29.1 | 33.1 |
> | MM-Eureka-7B-CPGD | 39.4 | 47.8 | 46.0 | 30.9 | 32.9 |

---

### Author Response · Authors · 2025-12-03
**Rebuttal Summary(1/2)**

To the newly assigned Area Chair,

Since reviewers can no longer join discussions or update scores, we sincerely thank them for their initial evaluations, which greatly helped improve our manuscript. We are especially grateful to the newly assigned Area Chair for taking on this responsibility. To support evaluation, we have included a “Rebuttal Summary” below, outlining the main revisions and responses addressing the reviewers’ concerns.

This benchmark fills a critical gap in evaluating Large Multimodal Models (LMMs) by focusing on fine-grained comparative reasoning over visually similar diagrams—a ubiquitous scenario in real-world education that challenges current SOTA models. Our work highlights multiple image-text misalignment as a primary bottleneck in current LMMs and also introduces three effective alignment strategies, comprising two training-free approaches and an alignment-oriented CoT fine-tuning method.

**Key strengths highlighted by reviewers include:**


* **Novel Benchmark Design**: All reviewers (ABAJ, 9Krr, bDac, 4GQo) emphasized that VisioMath identifies and addresses a significant, underexplored niche in multimodal mathematical reasoning, specifically the challenge of "figure-based options" with high visual similarity.

* **High-quality Data**: The reviewers praised the high quality of the dataset, noting the meticulous curation from real exams, the quantification of visual similarity, and the careful control of positional bias (ABAJ, 9Krr).

* **Deep Analysis**: The reviewers acknowledged our thorough analysis, which goes far beyond reporting accuracy. The innovative “option shuffling” strategy exposes positional heuristics, and failure categorization provides clear insights, revealing that the fundamental weakness of current LMMs lies in achieving multiple precise image–text alignment (9Krr, bDac, 4GQo).

* **Effective Improvement Strategies**: The proposed alignment strategies (layout consolidation, anchors, CoT finetuning) were recognized as effective and practical contributions (ABAJ, bDac, 4GQo).

---

> ### Author Response · Authors · 2025-12-03
> **Rebuttal Summary(2/2)**
>
> **In response to reviewer concerns, we provided the following clarifications and additional results:**
>
> * **Failure Case Analysis:** In response to requests for deeper analysis (4GQo, bDac), we have significantly expanded our qualitative analysis to include a detailed breakdown of vision recognition errors and specific failure cases in Appendix D.1.
>
> * **Expanded Model Coverage:** Addressing concerns regarding model scope (ABAJ), we evaluated 9 specialized math-oriented LMMs (e.g., MathCoder, VLM-R1). The results confirm that domain-specific pretraining alone fails to solve the fine-grained discrimination challenge. We present the complete results in Tables 7 and 8 (**50+ MLLMs**), with the three highest-performing models in Tables 2 and 3 for clarity.
>
> * **CoT Finetuning (Strategy 3) for Larger Models:**  To demonstrate the scalability of our approach (9Krr, bDac), we extended our fine-tuning strategy to larger models (Qwen2.5-VL-7B and 72B) and a distinct architecture (InternVL2.5). The results (Table 4) confirm that our alignment-oriented data consistently yields scalable performance gains.
>
> * **Benchmark Sourcing Bias:** Addressing concerns about potential geographical bias (9Krr, bDac), we collected and evaluated a supplementary dataset of international competition problems (e.g., AMC, Kangaroo Math). The performance trends are highly consistent with the main benchmark, confirming the universality of the visual reasoning patterns tested in VisioMath.
>
> * **Robustness of Similarity Metrics:**  Regarding the robustness of our similarity metric (9Krr), we justify using minimum pairwise similarity as a strict difficulty lower bound. Ablation studies (Appendix E.1) and Table 6 results confirm that quartile groupings and performance trends remain consistent across mean and median metrics. This statistical robustness verifies that edge cases do not meaningfully affect the overall dataset characterization.
>
> * **Ablation on the Embedding Model Used for Similarity Evaluation Metrics:**  Addressing the rationale for embedding model selection (bDac), we selected the Qwen encoder for its specialized diagrammatic training. Comparative retrieval experiments (Appendix E.2) demonstrate that unlike CLIP or BLIP, which prioritize generic semantics, Qwen embeddings accurately capture the fine-grained geometric topology essential for differentiating VisioMath options.
>
> * **Human Baseline:** Responding to the inquiry regarding human performance (4GQo), we conducted a full-scale human evaluation across the entire benchmark. As shown in Table 3, we can see that human performance moderately decreases as visual similarity among candidate diagrams increases, confirming that higher similarity introduces additional perceptual challenges. Second, beyond a certain similarity threshold, accuracy plateaus, suggesting that errors at this stage are driven more by conceptual understanding than by perceptual similarity. This suggests that while high similarity increases perceptual load, humans can still reliably distinguish fine-grained differences through careful observation.
>
> * **Additional Clarifications: We clarified and provided more analysis.**
>
>   1. The multiple-choice format is intentional to isolate comparative reasoning and prevent text-only shortcuts (ABAJ).
>
>   2. The effect of  Strategy 1 in single-image MLLMs and multi-image MLLMs is different (9Krr).
>
>   3. Strategy 2 benefits weaker models more than SOTA models with intrinsic spatial attention (9Krr).
>
>   4. MLLMs systematically rely on positional priors rather than random guessing under option shuffling (bDac).
>
>   5. The K-12 focus is justified by the transferability of diagram discrimination to other STEM domains (4GQo).
>
>   6. Distinguishing model types highlights architectural drivers of performance gaps (4GQo).
>
> The reviewers provided insightful feedback, primarily regarding experimental details and clarifications.  We believe the additional experiments and analyses have significantly enhanced the comprehensive nature and reliability of VisioMath.

---

### Meta-Review · Area_Chair_asN4 · 2026-01-07

**Summary:**

This paper introduces VisioMath, a benchmark targeting fine-grained comparative reasoning over visually similar diagrammatic answer options in K–12 mathematics.

The initial concerns are:

1) Scope, Generalization, and Benchmark Positioning:
Several reviewers questioned whether a K–12–focused benchmark sourced largely from Chinese exams could generalize to broader multimodal reasoning settings, and whether its contribution was sufficiently positioned relative to existing multimodal benchmarks (e.g., physics or scientific reasoning tasks).

2) Evaluation Depth and Human Baselines
Multiple reviewers raised concerns about the absence of a human baseline and requested deeper qualitative or fine-grained failure analysis beyond aggregate accuracy.

3) Generality of Proposed Improvement Strategies
Reviewers noted that the alignment-oriented CoT fine-tuning strategy (Strategy 3) was initially demonstrated on only one small model, limiting confidence in scalability and generality.

4) Presentation and Clarity
Reviewers also highlighted dense tables, confusing terminology, and insufficiently polished presentation, which initially detracted from accessibility.

**Reviewer Concerns:**

Human Evaluation is added. The analysis strengthened the paper’s central claim by showing that humans degrade much less sharply than LMMs under high visual similarity.

The authors significantly broadened model coverage, including specialized math-oriented LMMs and larger-scale models (Qwen2.5-VL 7B and 72B, InternVL2.5), demonstrating that observed failure modes and the gains from Strategy 3 are consistent across scales and architectures.

Presentation Improvements: Tables were simplified, wording was clarified, and less informative baselines (e.g., single-image models) were moved to the appendix, improving readability.

These concerns are addressed.

**Reviewer Scores:**

Domain Breadth: While the authors convincingly argue for transferability beyond K–12 mathematics, the benchmark remains domain-specific. Some reviewers may still view this as a limitation rather than a flaw.

Benchmark-Centric Contribution: The contribution remains more benchmark- and analysis-driven than algorithmically novel. The brought insights are limited despite recognizing its value.

---

### Decision · Program_Chairs · 2026-01-26

Accept (Poster)